



# The Dutch real-time gauge-adjusted radar precipitation product

Aart Overeem[1], Hidde Leijnse[1], Mats Veldhuizen[1], and Bastiaan Anker[1]

[1]R&D Observations and Data Technology, Royal Netherlands Meteorological Institute, Utrechtseweg 297, 3731 GA De Bilt, the Netherlands,

**Correspondence:** Aart Overeem (overeem@knmi.nl)

**Abstract.** The Dutch real-time gauge-adjusted radar quantitative precipitation estimation (QPE) product provides 5 min accumulations every 5 min on a $\sim 1\,\mathrm{km}^2$ grid covering the Netherlands and the area around it ($\sim 4.5 \times 10^5\,\mathrm{km}^2$). It plays a key role in hydrological decision-support systems and as input for nowcasts in order to inform decision makers. Major changes to the production of this QPE product were implemented on 31 January 2023, and include (polarimetric) fuzzy logic clutter removal, rain-induced attenuation correction and vertical profile of reflectivity correction. Moreover, the mean-field bias rain gauge adjustment was replaced by a spatially variable rain gauge adjustment. We evaluate the potential quality improvement resulting from these changes by comparing the last year of the old and the first year of the renewed QPE product. Clutter leading to overestimation in the old radar product is effectively removed in the renewed radar product. Evaluation against rain gauge accumulations shows a strong improvement. Average underestimation decreases by about ten percentage points to 15% over the Netherlands. Improvements of statistics are clear for daily precipitation over a large part of the QPE product domain, but also show the potential for incorporating rain gauge accumulations outside the Netherlands. The 1 h and daily extremes over the Netherlands are also better captured by the renewed product. Improvements in daily precipitation accumulations for the renewed product are stronger in the winter period than in the summer period. Finally, it is recommended to include Belgian and German rain gauge data in the product. The Dutch real-time 5 min gauge-adjusted precipitation radar dataset is publicly available at https://doi.org/10.21944/5c23-p429 (real-time) and https://doi.org/10.21944/e7zx-8a17 (archive) (KNMI Radar Team, 2018a, b).

## 1 Introduction

Accurate, timely and detailed precipitation information is essential for providing early warnings for extreme precipitation (e.g., for event organization, professionals working outside, and landslide risk), optimizing agriculture, and improving water management (e.g., flash flood forecasting). Rain gauges are accurate but cannot provide the needed coverage, whereas satellite precipitation products have limitations regarding accuracy and spatiotemporal resolution. Ground-based weather radar products do provide the needed space-time coverage. However, many sources of error can limit the usefulness of these products (Doviak and Zrnić, 1993; Fabry, 2015; Rauber and Nesbitt, 2018; Ryzhkov and Zrnic, 2019). To this end, real-time gauge-adjusted radar products optimally combine the accuracy of rain gauges with the coverage of radars.

Peer-reviewed literature providing an overview of the processing of nationwide (near) real-time radar precipitation products and their quality is relatively rare, especially that covering at least all seasons. There are however interesting studies on the





quality of real-time quantitative precipitation estimation (QPE) products. Several products cover the conterminous United States of America, such as the NCEP Stage IV radar product (evaluated over an 11 year period by Prat and Nelson, 2015; Nelson et al., 2016) and the Multi-Radar Multi-Sensor quantitative precipitation system (evaluated over case studies Zhang et al., 2016; Tang et al., 2020). For Europe, Park et al. (2019) describe a near real-time gauge-adjusted European radar precipitation product and evaluate over May–October for the years 2015–2017. For Switzerland, Germann et al. (2006) present an 8 year evaluation. For the Netherlands, Holleman (2007) provides a 6 year evaluation. For France, Figueras i Ventura and Tabary (2013) describe a single-polarization and two versions of dual-polarization (polarimetric) radar precipitation products, which are evaluated for 6 radars and a selection of events.

Here, we provide an overview and quality assessment for the Dutch real-time radar QPE product. The Royal Netherlands Meteorological Institute (KNMI) produces a real-time gauge-adjusted radar product of 5 min precipitation accumulations on a $\sim 1 \, km^2$ grid every 5 min, which is publicly available since 19 December 2018 (KNMI Radar Team, 2018a, b). This real-time QPE product covers the Netherlands and surrounding area ($\sim 4.5 \times 10^5 \, km^2$; Fig. 1a), and is currently produced with a latency of on average $\sim 2 \, min$. It is based on data from C-band ground-based weather radars from the Netherlands, Germany, and Belgium, and on data from 32 automatic rain gauges within the Netherlands. Major changes in the production of this QPE product have become operational as of 31 January 2023. These changes include dual-polarization non-meteorological echo removal (Overeem et al., 2020), rain-induced attenuation correction (Overeem et al., 2021), and vertical profile of reflectivity correction (Hazenberg et al., 2013). Quality information is derived in each processing step and subsequently used to generate a single surface reflectivity field from volume data for each radar. Further processing includes the statistical Gabella clutter filter (Gabella and Notarpietro, 2002), quality-based compositing of data from the contributing radars, conversion of reflectivity data to 5 min rainfall intensities using the Marshall-Palmer relationship (Marshall et al., 1955), and advection correction. Finally, a spatial adjustment factor field is computed employing 60 min rain gauge accumulations from a previous clock-hour (every hour on the hour). The adjustment factor field is applied to the most recent 5 min precipitation accumulation field.

The aim of this paper is twofold: first, to describe the datasets and algorithms used to produce this QPE product. Second, to evaluate the quality and limitations for the last year of the old product and for the first year of the renewed product. This is not only relevant for (potential) users of this dataset, but also quantifies the added value of the product renewal. Specific attention is given to the removal of non-meteorological echoes through evaluating maps with annual and maximum precipitation and relative frequency of exceedance. Evaluation is performed utilizing independent hourly and daily rain gauge accumulations by means of annual precipitation accumulation maps, scatter density plots, station-based spatial verification, and metrics as a function of gauge threshold value.

This paper continues with a radar and rain gauge data description (Sect. 2) and an overview of the methodology (Sect. 3). Next, an extensive evaluation of the old and renewed radar products against rain gauge accumulations is presented (Sect. 4), followed by a discussion (Sect. 5). Finally, conclusions and a research and development outlook for this product are provided (Sect. 7).





## 2 Data

The domain of the Dutch QPE product extends from $\sim 0°$E – $\sim 10°$E and from $\sim 49°$N to $\sim 56°$N, with the Netherlands in the center (Figure 1a). This region has a temperate, maritime climate. Two datasets are used: a full year before the transition to the new product (31 January 2022 – 31 January 2023; referred to as the old product), and a full year after this transition (31 January 2023 – 31 January 2024; referred to as the renewed product).

### 2.1 Radar data

The real-time radar product evaluated in this study is based on the 5 min volumetric (3D; i.e., a collection of typically 5–15 scans executed at different elevation angles) data from at most seven magnetron-based polarimetric C-band ground-based weather radars. Doppler clutter filtering is applied to both horizontal and vertical reflectivity factor ($Z_h$ and $Z_v$, respectively) data in order to remove non-meteorological echoes. Two radars are located in the Netherlands, two radars are located in Germany, 70 and three radars are located in Belgium. Figure 1a shows the locations of these radars including that of an additional radar in Germany (Neuheilenbach) that is used in the real-time product since 9 September 2024 (i.e., outside the evaluation period). On average 4.8 and 6.1 radars contribute to the real-time product for the old (31 January 2022 – 31 January 2023) and renewed (31 January 2023 – 31 January 2024) product, respectively. For both periods, the median number of contributing radars is 5. Over a large part of the radar product domain, the distance to the nearest radar is shorter than 150 km (Fig. 1a). Notable are the 75 longer distances to the nearest radar in the Dutch-German border region. More information on the Belgian radars is provided by Goudenhoofdt and Delobbe (2016), on the Dutch radars by Beekhuis and Mathijssen (2018) and on the German radars by Werner (2014).

Radar data processing is described in Sect. 3. The end result is a 2D real-time radar product of 5 min accumulations every 5 min on a $\sim 1$ km $\times$ $\sim 1$ km grid with almost full data availability. The 5 min accumulations are aggregated to 1 h (every hour 80 on the hour) and daily (ending 0 UTC, 6 UTC, or 8 UTC) accumulations for comparison against KNMI automatic rain gauge accumulations from the Netherlands (1 h) and against daily rain gauge accumulations from different networks across the entire radar product domain. Accumulations were only computed when data availability is at least 83.3 %. Annual accumulations were generated without this data availability criterion.

The combined daily radar–gauge availability is generally 95 % or higher (Fig. 1b–c). Lower availability is caused by missing 85 gauge records, except for the renewed radar product for the (south)western and southeastern part of the product domain. This extended coverage with lower availability can be attributed to the Wideumont radar in the southeast that started contributing in August 2023 and the addition of longer range scan data from the Jabbeke radar in the southwest.



## 2.2 Rain gauge data

### 2.2.1 KNMI 10 min and hourly automatic rain gauge data

KNMI operates an automatic rain gauge network that electronically measures precipitation accumulations based on the displacement of a float placed in a reservoir (KNMI, 2000). The locations of these 32 gauges, with a density of ∼1 gauge per 1000 km$^2$ over the land surface of the Netherlands, are displayed in Fig. 1a. The unvalidated 10 min data were aggregated to 1 h precipitation accumulations (every hour on the hour). Only these accumulations were used to compute a spatial adjustment factor field for real-time radar product adjustment. In addition, validated 1 h gauge accumulations (every hour on the hour) were obtained for evaluation purposes. Since the automatic gauge accumulations from a previous clock-hour (every hour on the hour) are employed in the adjustment, the evaluation is considered to be independent.

### 2.2.2 KNMI daily manual rain gauge data

The manual rain gauge network operated by KNMI provides daily precipitation accumulations (end time of observation is 08:00 UTC) from 317 (old product) and 319 (renewed product) gauges (density of ∼1 gauge per 100 km$^2$). Volunteers empty these rain gauges at 8 UTC and pass on the readings to KNMI (KNMI, 2000). Here, the accumulations that have been validated by KNMI staff are employed for evaluation purposes. For qualitative comparison purposes, a product with spatially interpolated daily rain gauge accumulations at a 1 km × 1 km grid over the Netherlands is used (Soenario et al., 2010; Siegmund, 2014).

### 2.2.3 DWD daily rain gauge data

Daily manual and automatic rain gauge accumulations over Germany were obtained on 6 December 2024 from the DWD open data portal CDC (DWD, 2024); the "historical", more quality controlled data were available through 31 December 2023, and as of 1 January 2024 the "recent" less quality controlled data were employed. These accumulations are utilized for the spatial evaluation for 468 locations (old radar product) and 649 locations (renewed radar product). The end time of observation is 06:00 UTC.

### 2.2.4 ECA&D daily rain gauge data from Belgium, France, Luxembourg and the United Kingdom

The non-blended daily precipitation accumulations were obtained on 20 December 2024 from the European Climate Assessment and Dataset (ECA&D, https://www.ecad.eu, last access: 23 January 2025) project (Klein Tank and coauthors, 2002; Klok and Klein Tank, 2008) and are used for the spatial evaluation. This concerns rain gauge data from 170–171 locations in Belgium (end time of observation 00:00 UTC) and Luxembourg (end time of observation 00:00 UTC), 50–64 locations in France (end time of observation 08:00 UTC), and 2–8 locations in the United Kingdom (end time of observation 08:00 UTC). This includes non-downloadable series (i.e., included in ECA&D for production of derived data but only accessible through the data owner). These data have been quality controlled by the ECA&D team (Project team ECA&D, Royal Netherlands Meteorolog-





ical Institute KNMI, 2021) and often also by the party that has delivered the data. Instead of ECA&D, German rain gauge data were retrieved from their original source, the DWD open data portal.

For qualitative comparison purposes, a spatially interpolated daily ECA&D rain gauge accumulation product over Europe
is used (E-OBS version 30.0e Cornes et al., 2018; Copernicus Climate Change Service, 2024). This dataset was aggregated to annual precipitation accumulations.

## 3   QPE production

Production of the QPE product involves several processing steps, starting with steps that are applied for each radar separately. This is followed by processing steps that are done on the combined product. These two collections of steps are shown as flow
charts in Fig. 2, ultimately leading to the renewed real-time radar product. The production process for the old radar product lacks most of these processing steps and is summarized in the next subsection.

### 3.1   Processing old radar product

For each radar, a 2D radar pseudo-constant-altitude plan position indicator (pseudoCAPPI) of $Z_\mathrm{h}$ is produced by logarithmic averaging of the $Z_\mathrm{h}$ (dB$Z_\mathrm{h}$) data from three scans. This is converted to precipitation intensities $R$ through a $Z_\mathrm{h} = 200R^{1.6}$
relationship (Marshall et al., 1955). A constant mean-field bias adjustment factor is subsequently applied to this intensity field (Holleman, 2007). The mean field bias adjustment factor is based on radar and rain gauge 1 h accumulations from a previous clock-hour (every hour on the hour; as explained in subsect. 3.9). The resulting intensity fields from each radar are transformed to 5 min accumulations and subsequently combined into a single 2D composite via range-weighted compositing.

### 3.2   Fuzzy logic clutter removal

The processing for the renewed radar product starts with the removal of non-meteorological echoes (clutter) by employing the fuzzy logic algorithm from the wradlib open-source Python library for weather radar data processing (Heistermann et al., 2013; Overeem et al., 2020; Mühlbauer et al., 2020). Clutter results from reflections of non-meteorological targets, such as the ground, buildings, trees, cars, sea surface, ships, birds, and airplanes. This may be exacerbated by anomalous propagation leading to the beam hitting the ground or sea surface. Interference by other sources of radio waves such as wireless networks
and the sun also cause clutter (Gourley et al., 2007; Fabry, 2015; Huuskonen et al., 2016; Saltikoff et al., 2016). Removal of clutter-affected data is important, since it can lead to severe precipitation overestimation.

The fuzzy logic algorithm uses decision variables to classify clutter for each voxel from each scan separately (Vulpiani et al., 2012; Crisologo et al., 2014; Overeem et al., 2020). The following decision variables are employed for the Dutch radar data: texture (spatial variability) of the differential reflectivity ($Z_\mathrm{DR}$), the copolar correlation coefficient ($\rho_\mathrm{HV}$), texture of $\rho_\mathrm{HV}$,
depolarization ratio (Ryzhkov et al., 2017), and clutter phase alignment (Hubbert et al., 2009b, a). For each decision variable, the degree of membership to the meteorological target class is calculated employing a membership function. A weighted average of the degree of membership to the meteorological target class is computed using all decision variables, each having its





own weight. A voxel is classified as clutter if this weighted average is below a threshold value of 0.6. In that case the associated $Z_h$ is set to the "nodata" value (i.e., areas void of data). All details regarding the fuzzy logic algorithm and its (parameter)
settings are provided by Overeem et al. (2020).

The Belgian radar data lack the decision variable clutter phase alignment. Utilizing the same membership functions, only three decision variables are used and with a different weight: texture of $Z_{DR}$ (0.5), texture of the two-way differential propagation phase (0.3), and $\rho_{HV}$ (0.2). Moreover, the employed threshold value in the classification is 0.3.

Goudenhoofdt and Delobbe (2016) give more information on the Belgian radar processing. For the German radars, no
polarimetric data were publicly available for the periods that are used in this paper. However, DWD has already applied several post-processing algorithms, including a (partly polarimetric) fuzzy logic clutter detection and removal algorithm for the two 1 year periods. Werner (2014) provide details on the German radar processing.

### 3.3   Attenuation correction

Rain-induced attenuation can lead to severe precipitation underestimation for C-band weather radars (Hitschfeld and Bordan,
1954; Tabary et al., 2009; Fabry, 2015). The two-way path-integrated attenuation (PIA) along the radar beam can be estimated and corrected for. Here, the recipe from Overeem et al. (2021) is followed. Attenuation is computed from specific differential phase ($K_{dp}$) (Bringi et al., 1990) assuming a linear relation between the two for the Belgian and Dutch radars. Because real-time access to dual-pol data from the German radars was not possible, a Hitschfeld-Bordan type of algorithm was applied there. This method, called the Modified Kraemer method, uses the method proposed by Hitschfeld and Bordan (1954) with an
additional limit on the attenuation correction of 10 dB and on the resulting reflectivity of 59 dB$Z_h$ to avoid instability of the algorithm (Jacobi and Heistermann, 2016; Overeem et al., 2021). The final step for both methods is to increase the $Z_h$ value for each voxel by the computed attenuation. The method using $K_{dp}$ was shown to outperform the Modified Kraemer method using 1 year of data from the two C-band radars in the Netherlands, but the stability of the Modified Kraemer method was also confirmed (Overeem et al., 2021).

For both methods, the wradlib open-source Python library for weather radar data processing (Heistermann et al., 2013; Mühlbauer et al., 2020) is utilized. Since the correction is meant for attenuation caused by liquid precipitation, a voxel is only used to compute the attenuation in case its height is below the forecasted freezing-level height from the numerical weather prediction model HARMONIE-AROME. And for the method using $K_{dp}$, voxels classified as clutter are not employed to compute attenuation (Wradlib, 2020c). Moreover, attenuation correction is only applied to voxels that are not classified as
clutter according to the fuzzy logic algorithm (for the Belgian and Dutch radar data). Note that attenuation correction is applied to voxels above the freezing-level height.

For the German radars, for which no polarimetric data were available, the Modified Kraemer method is applied per elevation scan via the $k_h = \alpha Z_h^{\beta}$ relation. Values of $\alpha$ and $\beta$ are reduced from their initial values in an iterative procedure until the constraints $Z_{h,cor} \leq 59$ dB$Z_h$ and PIA $\leq 10$ dB are met. In case constraints are not met, no attenuation correction is performed.
More details are provided by Jacobi and Heistermann (2016); Overeem et al. (2021); Wradlib (2020b).



Here, the algorithm of Overeem et al. (2021) is extended to attenuation correction for vertically polarized reflectivity employing $k_v = 0.058978K_{dp}$, in order to obtain the attenuation-corrected $Z_{DR}$. This is important, since it is used in the vertical profile of reflectivity correction.

### 3.4 Vertical profile of reflectivity correction

Precipitation generally has a non-uniform vertical structure. Therefore, the increased height of the radar sampling volume at long range can lead to errors in surface precipitation estimates. To address this, the vertical profile of reflectivity (VPR) correction is applied as suggested by Hazenberg et al. (2013, 2014), extended with the polarimetric melting layer detection from Boodoo et al. (2010). The algorithm estimates two idealized VPR profiles, one for stratiform, and one for undefined precipitation based on all data that has been classified as one of these two precipitation types. Note that no VPR estimation or
correction is applied for convective precipitation, because there is much more vertical mixing in convection. The VPR estimates include an uncertainty estimate that is used in subsequent steps of the QPE processing chain. The VPR estimates are then used to extrapolate all reflectivity data down to the ground. For both estimation and correction for VPR, the shape of the beam is assumed to be Gaussian.

### 3.5 Quality-based merging of radar scans

We use quality information to combine data from several scans to derive a 2D surface reflectivity product. Appendix A details the estimation of a quality index $Q_T$ for each voxel. The $N$ voxels that contribute to a given pixel in the 2D surface reflectivity product are combined as follows into a quality-weighted reflectivity ($Z_{h,Q}$):

$$Z_{h,Q} = \frac{\sum\limits_{j=1}^{N} Q_{T,j} Z_{h,j}}{\sum\limits_{j=1}^{N} Q_{T,j}}, \tag{1}$$

where both reflectivities are expressed in mm$^6$ m$^{-3}$.

A combined quality index is also computed for each pixel in the 2D radar product ($Q_r$). The basic principles are that each added voxel $j$ raises the value of the quality index, and that the resulting quality index ranges between 0 and 1:

$$Q_r = 1 - \prod_{j=1}^{N} (1 - Q_{T,j}). \tag{2}$$

The advantage of this approach is that voxels from scans that have non-zero quality can compensate for the zero quality of voxels from other scans, and that voxels with lower quality have a much lower weight in the final product.

### 3.6 Gabella clutter filter

The Gabella clutter filter from the wradlib open-source Python library for weather radar data processing (Gabella and Notarpietro, 2002; Heistermann et al., 2013; Mühlbauer et al., 2020; Wradlib, 2020a) has been successfully applied for detection and





removal of residual clutter (e.g., RADKLIM for Germany and EURADCLIM for Europe; Lengfeld et al., 2019; Overeem et al., 2023a). Here, it is applied to the 2D Cartesian $Z_{h,Q}$ data from each radar and time interval separately. First, large gradients of $Z_{h,Q}$ in space are detected. For each central pixel, the number of pixels within a square lattice of $5 \times 5$ pixels is counted that have a $Z_{h,Q}$ value less than $6\,\text{dB}$ lower than the central pixel. The central pixel is classified as clutter if this number of pixels is fewer than 6. Second, the ratio between the area and circumference for contiguous echo areas with $Z_{h,Q}$ above $0\,\text{dB}Z_h$ is calculated. The central pixel is classified as clutter if this ratio is lower than 1.3 pixels. The central pixels classified as clutter do not contribute to the radar product by setting the $Z_h$ to the "nodata" value and the quality index to 0.

## 3.7 Compositing of reflectivity images

The radar quality index (Appendix A) is also used to weigh the reflectivity values from individual radars into a combined composite of $Z_{h,Q}$. It employs the same algorithms as for the construction of 2D surface reflectivity products for each radar: the quality weighted $Z_h$ is computed using Equation 1 and the corresponding quality index is computed using Equation 2.

## 3.8 Conversion to rainfall intensities and advection correction

Reflectivities $Z_{h,Q}$ ($\text{mm}^6\,\text{m}^{-3}$) are converted to rainfall intensities $R$ ($\text{mm}\,\text{h}^{-1}$) employing the $Z_{h,Q}$–$R$ relationship (Marshall et al., 1955):

$$Z_{h,Q} = 200R^{1.6}. \tag{3}$$

This implicitly assumes that precipitation is liquid, i.e., rain.

Storms that move more than 1 km (the spatial resolution of the QPE product) in 5 min (its temporal resolution) will lead to artifacts in precipitation accumulations. This is corrected by interpolating radar composites in time. For this purpose, advection vectors are computed using the implementation of Farnebäck (2003) in the OpenCV library (http://opencv.org) (last access: 4 February 2025). These vectors are then used to interpolate the radar composites in time. The resulting interpolated precipitation intensity becomes a weighted average of the previous ($t = t_0$) and current ($t = t_1$) precipitation composite:

$$
\begin{aligned}
R_{\text{int}}(\boldsymbol{x},t) = &\frac{t_1 - t}{t_1 - t_0} R(\boldsymbol{x} + \boldsymbol{v}(t - t_0), t_0) + \\
&\frac{t - t_0}{t_1 - t_0} R(\boldsymbol{x} - \boldsymbol{v}(t_1 - t), t_1),
\end{aligned} \tag{4}
$$

with $t$ the time ($t_0 \le t \le t_1$), $R_{\text{int}}(\boldsymbol{x},t)$ the interpolated precipitation intensity, $R(\boldsymbol{x},t)$ the composite precipitation intensity, $\boldsymbol{x}$ is the location of the pixel, and $\boldsymbol{v}$ is the advection vector ($\text{m}\,\text{s}^{-1}$). Subsequently, the advection-corrected 5 min precipitation accumulation is computed by aggregating a number of interpolated composites and multiplying by the time interval between the different interpolated composites:

$$R_{\text{acc}}(x, t_1) = \frac{t_1 - t_0}{N + 1} \sum_{i=0}^{N} R_{\text{int}}\left(x, t_0 + i\frac{t_1 - t_0}{N}\right), \tag{5}$$





where $N - 1$ is the number of composites that is generated between the previous ($t_0$) and present ($t_1$) composite. Since the horizontal velocity of showers is expected to be lower than $50 \text{ m s}^{-1}$, $N = 14$ is used for this spatiotemporal resolution (1 km and 5 min).

### 3.9 Gauge adjustment of radar accumulations

The method for adjustment of radar accumulations with rain gauge accumulations is a modified version of the Barnes' objective

analysis (Barnes, 1964). The aim is to have a method that is robust in situations with large precipitation gradients and limited gauge density, and that can use quality information of both the radar and rain gauge precipitation estimates. The spatial adjustment factor for a given pixel is computed by the ratio of distance-weighted sums of the unadjusted radar and corresponding rain gauge precipitation accumulations at the gauge locations. The computation of the spatial adjustment factor field is described in Appendix B. Although the adjustment factor field itself is computed based on radar and gauge accumulations from the

same 60 min interval, it does not encompass the radar data from the current 5 min time interval due to 50 min latency of gauge data, that were only available every clock-hour (every hour on the hour). Hence, the adjustment factor field from a previous clock-hour (every hour on the hour) was used. For instance, from 10:50–11:50 UTC the 5 min accumulations were adjusted employing the spatial adjustment factor field based on 60 min radar and gauge accumulations from 09:00–10:00 UTC. As of 11:50 UTC, the applied spatial adjustment factor field was computed based on data from 10:00-11:00 UTC. And possibly,

sometimes even an older adjustment factor field was employed.

## 4 Evaluation results

The performance of the old radar product in its last year is compared to the performance of the renewed radar product in its first year. Performance evaluation is done by qualitative analysis of maps of accumulations and exceedance probabilities as well as by quantitative comparison against rain gauge accumulations. Statistics are presented as a function of gauge accumulation

threshold value, geographic location, and season.

Radar precipitation accumulations from pixels directly above a rain gauge ($R_{\text{radars}}$) are evaluated against the corresponding rain gauge precipitation accumulations ($R_{\text{gauges}}$). Residuals are defined as the radar accumulations minus the corresponding gauge accumulations. The following metrics are computed:

  – The relative bias is the mean of the residuals divided by the mean of the gauge accumulations. It is expressed as a

percentage:

$$\text{Rel.bias} = \frac{\bar{R}_{\text{res}}}{\bar{R}_{\text{gauges}}} \times 100 = \frac{\sum\limits_{i=1}^{n} R_{\text{res},i}}{\sum\limits_{i=1}^{n} R_{\text{gauges},i}} \times 100, \tag{6}$$

with $n$ the total number of radar-gauge pairs.





– The coefficient of variation of the residuals ($\mathrm{CV_{res}}$, called CV in this paper) is computed as the standard deviation of the residuals, divided by the mean of the gauge accumulations:

$$\mathrm{CV_{res}} = \frac{\sqrt{\frac{1}{n-1}\sum\limits_{i=1}^{n}\left(R_{\mathrm{res},i} - \bar{R}_{\mathrm{res}}\right)^2}}{\bar{R}_{\mathrm{gauges}}}. \tag{7}$$

This is a measure of spread.

– The Pearson correlation coefficient ($\rho$) or its squared value ($\rho^2$) between radar and gauge accumulations is computed:

$$\rho = \frac{\mathrm{cov}\left(R_{\mathrm{gauges}}, R_{\mathrm{radars}}\right)}{s\left(R_{\mathrm{gauges}}\right) \times s\left(R_{\mathrm{radars}}\right)}, \tag{8}$$

with $s$ the sample standard deviation.

– The Kling-Gupta Efficiency (KGE) is computed to summarize performance in one score (Kling et al., 2012):

$$KGE = 1 - \sqrt{(\rho - 1)^2 + (\beta - 1)^2 + (\gamma - 1)^2}, \tag{9}$$

with bias ratio $\beta$ (being equal to the rel. bias + 1):

$$\beta = \frac{\bar{R}_{\mathrm{radars}}}{\bar{R}_{\mathrm{gauges}}}. \tag{10}$$

The variability ratio $\gamma$ is given by:

$$\gamma = \frac{\mathrm{CV_{radar}}}{\mathrm{CV_{gauge}}} = \frac{s_{\mathrm{radar}}/\bar{R}_{\mathrm{radar}}}{s_{\mathrm{gauge}}/\bar{R}_{\mathrm{gauge}}}. \tag{11}$$

Relative frequencies of exceedance are computed for each pixel separately and obtained by dividing the number of exceedances by the number of values with data (i.e., not equal to the "nodata" value, denoting outside the radar product domain). Missing radar files are not taken into account in the computation of the number of exceedances and the number of values with data.

## 4.1 Qualitative analyses

Maps of the maximum 5 min precipitation accumulation over 1 year as shown in Fig. 3a,b reveal values of 10 mm and higher. These are mostly associated with shipping tracks and wind farms over the North Sea and the English Channel, and metal cranes and containers of the Port of Rotterdam ("Maasvlakte (2)") for the old radar product (Fig. 3a). They are hardly present in the renewed radar product, showing the effectiveness of the fuzzy logic algorithm and the Gabella clutter filter, with only some remaining isolated spots that are probably due to wind farms (Fig. 3b). Clutter for the land surface seems less of an issue for the old radar product, and even less so for the renewed radar product. Maps of the relative frequency of 5 min precipitation $\geq 0.01\,\mathrm{mm\,h^{-1}}$ and $>6\,\mathrm{mm\,h^{-1}}$ for the old radar product (Fig. 3c,e) reveal clutter for "Maasvlakte (2)", shipping tracks (much more pronounced for $6\,\mathrm{mm\,h^{-1}}$) and wind farms in the North Sea, and quite some suspicious areas for the land surface that are





likely clutter. Few clutter areas appear in the renewed radar product, except for the radial patterns over the North Sea and the
English Channel for $\geq 0.01\ \mathrm{mm\ h^{-1}}$ pointing toward the Belgian radar in Jabbeke, which are caused by interference (Fig. 3d).

Figure 3 shows that to some extent maximum accumulations, but especially relative frequencies of exceeding a threshold,
are generally larger for the renewed radar product. This will be partly due to the much wetter 1 year period, but also indicates
lower underestimation due to improved processing. Note the high 5 min maxima in the southeastern part for the old radar
product, which seem connected to rain showers. The spatial patterns here are typical for storms that move faster than a single
pixel (1 km) in the time between two samples (5 min). The renewed product (Fig. 3b) does not exhibit such patterns, indicating
that the advection correction (see Sect. 3.8) is effective.

The annual precipitation maps (Fig. 4a,c) reveal all types of artifacts found in Fig. 3, confirming that clutter is a large issue
for the old radar product, whereas mainly limited to a single spoke-like artifact caused by interference for the renewed radar
product, resulting in accumulations of 1420 mm or higher.

## 4.2   Annual precipitation maps

Figure 4 shows the annual precipitation maps for the old and renewed radar products and the corresponding E-OBS inter-
polated rain gauge product. The old radar product severely underestimates precipitation, especially outside the Netherlands,
where no gauge accumulations have been employed in the adjustment. Underestimation is less severe for the renewed radar
product. Precipitation gradients over the Netherlands roughly coincide with the E-OBS gauge product. The large area with high
precipitation accumulations in Germany, that is missed in the old radar product, is captured by the renewed radar product. This
likely shows the added value of the VPR correction, the rain-induced attenuation correction, the contribution of the Borkum
radar, and the much higher weight of data from the nearest radar in the compositing.

The effect of severe beam blockage can be noticed in the east and northeast of the Netherlands for both products, pointing
towards the Dutch Herwijnen radar (see Fig. 1). For the old product, beam blockage becomes apparent only at longer range
from the radar, where the precipitation estimation is mainly based on the data from the Herwijnen radar (other radars have much
more weight close to the radar). The severe beam blockage in easterly direction from the Herwijnen radar, mainly caused by a
line of trees, underpins the need for radar site protection. The effects of beam blockage over the northeast of the Netherlands,
mainly caused by orography by the Utrecht Hill Ridge (Overeem et al., 2023b), is decreased for the renewed radar product due
to the mitigating effect of the addition of data from the German radar in Borkum (Fig. 1a).

## 4.3   Scatter density plots

Scatter density plots and metrics of radar versus rain gauge precipitation accumulations are employed to assess the overall
performance of the radar product for the Netherlands. For both 1 h and daily precipitation accumulations, the renewed radar
product performs better than the old radar product (Fig. 5). Notably, the underestimation decreases from $\sim 17\%$ to $\sim 7\%$
for hourly and from $\sim 24\%$ to $\sim 15\%$ for daily accumulations. But also the values for CV are lower and values for $\rho^2$ and
KGE are higher for the renewed radar product. The improvements for the renewed radar product are more apparent in the
daily scatter density plots than in the hourly plots through better alignment along the $1:1$ line. The fact that underestimation





by radar is lower in magnitude compared to automatic rain gauges than in comparison to manual rain gauges, is consistent with the finding reported in Brandsma (2014), namely that automatic rain gauges are biased low by 5-8% compared to manual rain gauges. The underestimation for true precipitation events in the old radar product is expected to be larger, because of
compensation by clutter leading to overall less underestimation. This is confirmed by the many outliers for (near) zero daily gauge accumulations (and a few outliers for 1 h accumulations), that are not visible anymore in the renewed radar product. Hence, the improvement in the bias may be even larger.

### 4.4 Summer storm Poly

An example of the capability of the renewed radar product for real-time precipitation monitoring is shown in Fig. 6. Summer
storm Poly struck the Netherlands 5 July 2023 with locally more than 25 mm in 24 h (Fig. 6a–b). Daily precipitation patterns and accumulations from the radar product match quite well with those from an interpolated rain gauge product, based on validated data from 319 rain gauges from the Netherlands. Only the radar product is available in real-time and provides much more spatial detail and can hence capture the counterclockwise rotational movement of the low pressure area (Supplement (Movie S1)). The scatter density plot shows a fairly high correspondence of the daily radar accumulations with the manual rain
gauge accumulations from the Netherlands. It is expected that the old product performs less well primarily because of the lack of attenuation correction.

### 4.5 Spatial evaluation

The performance of daily radar precipitation accumulations is quantified for 1008 (old radar product) and 1210 (renewed radar product) rain gauge locations employing independent daily gauge accumulations from networks in Belgium, Germany,
Luxembourg, the Netherlands, and the United Kingdom at their default measurement interval (Fig. 7; note that this does not include the KNMI automatic gauges used for adjustment). Underestimation is severe for the old radar product (Fig. 7a), especially when the distance to the nearest employed automatic gauge from the Netherlands and to the nearest radar is long (Fig. 1a). The renewed radar product displays less underestimation over the land surface of the Netherlands, southwest Belgium and northwestern Germany (Fig. 7b). The VPR correction, the rain-induced attenuation correction, and the much higher weight
of the nearest radar in the compositing, likely play an important role here, as well as the addition of the German radar Borkum in the northeast of the radar product domain. Notable is the overestimation for more gauge locations in Germany for the renewed radar product. The relative bias in parts of Germany is even closer to zero than in the Netherlands. Perhaps this is also related to different radar hardware calibration. Another explanation is that the German Weather Service has already corrected their radar data for attenuation, so it has been corrected for attenuation twice (although this also holds for the old product). This indicates
the importance of provenance in radar metadata. As for the annual precipitation accumulations shown in Fig. 4, the influence of the severe beam blockage to the east of the Herwijnen radar is apparent in the relative bias.

Compared to the old radar product, the values for $\rho$ generally increase (Fig. 7c–d) and the values for CV generally decrease for the renewed radar product (Fig. 7e–f), confirming the improved performance for precipitation estimation.



## 4.6 Extremes

The results shown so far are based on all available data, i.e., without thresholding. Here, the quality of radar precipitation accumulations for a range of rain gauge precipitation threshold values is examined. Figures 8 and 9 show the values for relative bias, $\rho$ and CV, and the number of radar-gauge pairs as a function of this threshold value (step size of 1 mm) for both products for 1 h accumulations and daily accumulations, respectively.

The underestimation becomes generally larger for increasing gauge threshold values, but is less severe for the renewed radar product and fairly constant in the 5–15 mm threshold range (Fig. 8a) or the entire threshold range (Fig. 9a) for the renewed product. This is possibly caused by the attenuation correction. Underestimation is only fairly constant for the old product for daily threshold values up to $\sim 20$ mm. The value for $\rho$ usually decreases with increasing threshold value, with the renewed product performing best (Figs. 8b and 9b), except for 1 h threshold values above $\sim 13$ mm, for which erratic behavior is found, likely related to small sample size (Fig. 8d). The value for CV only strongly decreases from 0 to 1 mm, and remains fairly stable for larger threshold values. Performance is similar for the old and renewed radar products for 1 h threshold values ranging from 0 mm to $\sim 10$ mm, and slightly better performance for the old product for threshold values larger than $\sim 10$ mm (Fig. 8c). For higher gauge threshold values, the old product outperforms the renewed product. This is possibly related to the fact that both products cover a different 1 year period. In contrast, for daily accumulations, the value for CV generally decreases with better performance for the renewed product for threshold values ranging from 0 mm to $\sim 25$ mm, and similar or slightly better performance for the old radar product for the larger threshold values (Fig. 9c). Finally, the number of radar-gauge pairs is generally (slightly) higher for the renewed product and comparable or lower for the highest threshold values (Figs. 8d and 9d).

## 4.7 Seasonal dependence

To study a possible seasonal dependence of quality of radar precipitation accumulations, Fig. 10 shows scatter density plots of daily accumulations (see also Fig. 5c–d) for 5 month winter (Nov–Mar) and summer (May–Sep) periods. The following conclusions are drawn:

- The precipitation underestimation is clearly higher in the winter period compared to the summer period, with a much larger difference of $\sim 15$ percentage points for the old radar product compared to the $\sim 8$ percentage points for the renewed radar product (Fig. 10).

- For both products, in the summer period, values for $\rho$ are (slightly) worse and values for CV are worse compared to the values for the winter period, which may be related to the expected larger representativeness errors between radars and gauges in case of more local, convective rainfall.

- Improvements in relative bias, $\rho^2$, and KGE for the renewed radar product with respect to the old radar product are more pronounced in the winter period. This may indicate the effectiveness of the VPR correction and the fact that the renewed product uses data from closer to the ground, partly due to more radars, and especially close to a radar.



## 5 Discussion

The 1 year evaluation periods for the old and renewed radar products do not overlap. Hence, different environmental conditions and precipitation characteristics between those years may also have influenced the comparison between the old and renewed radar products. It is assumed that the effect of this is relatively minor for most of the comparisons presented in this paper
because of the lengths of the periods and the resulting large number of samples. The largest effect of using different periods is expected to be on the performance statistics of more extreme precipitation as presented in Figs 8–9. This holds especially for high precipitation thresholds, as these are associated with a much lower number of samples (see panels d of these figures).

The latency of rain gauge data currently prohibits adjustment of radar with gauge accumulations from the same time interval (see Sect. 3.9). Computing a spatial adjustment factor field on past data is not uncommon and useful (Park et al., 2019; Imhoff
et al., 2021), and can be the only option. However, this not only results in a less representative adjustment factor field for the current 5 min time interval, but can also cause a sudden change in the 5 min radar precipitation accumulations due to the change of adjustment factor field once an hour (instead of a more gradual change when the adjustment factor field would be computed every 5 min). This is clearly visible in the Supplement (Movie S1) for 04:50–04:55 UTC, 05:50–05:55 UTC and 06:50–06:55 UTC. The advantage of the fact that both products use slightly older rain gauge data for this study is that comparisons of the
products against 1 h automatic gauge data are fully independent.

An obvious improvement to the QPE product would be to apply real-time gauge accumulations for the adjustment instead of delayed gauge data as is the case for the renewed product. This potential improvement is assessed by rerunning the adjustment for the renewed radar product. For this experiment, the spatial adjustment factor field is computed and applied based on the unadjusted 1 h radar accumulations from the same hour. This mimics the quality of a product for which gauge accumulations
without latency would have been available for adjustment. Assessing the resulting daily accumulations leads to an overall improvement with respect to the renewed radar product (Fig. 5d): less underestimation ($\sim 12\%$ versus $\sim 15\%$), higher values for $\rho^2$ (0.90 versus 0.86), lower values for CV (0.56 versus 0.64), and hence higher values for KGE (0.87 versus 0.83).

There are several remaining (spatially variable) sources of error that have not been corrected for, such as wet radome attenuation (Germann, 1999; Kurri and Huuskonen, 2008), beam blockage, hardware calibration errors (Frech et al., 2017), and
assuming a fixed $Z_\mathrm{h} - R$ relationship (Marshall et al., 1955; Uijlenhoet, 2001, 2008). These, and the fact that the assumptions underlying the correction algorithms that have been applies are not always valid, contribute to deviations (also spatially variable) found in the radar precipitation product. The relatively low network density of automatic rain gauges employed in the adjustment and the fact that no gauge data are employed outside the Netherlands, limit the effectiveness to counteract these sources of error. The use of the threshold $T$ in the radar-gauge adjustment (Appendix B) is meant to prevent small absolute
differences in radar or gauge accumulations leading to large differences in the spatial adjustment factor field. However, this may also lead to less or no adjustment for low radar precipitation accumulations, hence not compensating for underestimation. So, for regions and intervals where precipitation is not captured by the relatively small number of gauges, the precipitation underestimation by radar cannot be compensated for. Developments and recommendations to address sources of error in the real-time radar precipitation product are provided in Sect. 7.2.



Since the evaluation is performed against rain gauges, part of the differences can also be attributed to representativeness errors between radars measuring aloft in a large volume (typically $\sim 1\,\mathrm{km}^3$) and gauges measuring locally at the Earth's surface (Kitchen and Blackall, 1992) with a catchment area of typically only $\sim 500\,\mathrm{cm}^2$. This is especially true for short durations (e.g., 1 h) where the limited time integration provides only limited compensation for the differences in volumes. Another part of the differences between radar and gauges is caused by sources of error in gauge precipitation estimation (WMO, 2023).

## 425   6   Data availability

The 5 min real-time precipitation radar dataset is available from the KNMI Data Platform at https://doi.org/10.21944/5c23-p429 (real-time) and https://doi.org/10.21944/e7zx-8a17 (archive) (KNMI Radar Team, 2018a, b) in KNMI HDF5 format containing metadata, such as geographical information (Mathijssen et al., 2019). The data are provided in real time with one file every 5 min. In this study, the archive has been accessed, which provides a tar file for each day. The data are in

UTC. Object "/image1/image_data" contains the 5 min precipitation accumulation (mm). For the renewed radar product, two additional objects are available: object "/image2/image_data" contains the quality index, and object "/image3/image_data" contains the adjustment factor field (dB). Note that "/imageN_data" objects are scaled, which can be found in the metadata ("/imageN/calibration/calibration_formulas"). For instance, the precipitation accumulations need to be multiplied by 100 to obtain the precipitation in millimeters since they are stored as integers in hundredths of millimeters (0.01 mm resolution). The

"/image1/calibration_out_of_image" value in "image1/image_data" is used to denote areas outside the radar product domain for that particular time interval, i.e., a "nodata" value (areas void of data).

## 7   Conclusions and outlook

### 7.1   Conclusions

The Dutch real-time gauge-adjusted radar QPE product is available since 19 December 2018 and provides 5 min accumulations

every 5 min on a $\sim 1\,\mathrm{km}^2$ grid covering the Netherlands and the area around it. Major changes in processing became operational on 31 January 2023, such as clutter removal through a fuzzy logic algorithm (Overeem et al., 2020) and rain-induced attenuation correction (Overeem et al., 2021) on Belgian and Dutch radar data, vertical profile of reflectivity (VPR) correction (Hazenberg et al., 2013) on all radar data, quality-based merging of radar reflectivities from different scans and radars, and a spatially variable instead of a mean-field bias adjustment with rain gauge accumulations. The last year of the old dataset and the first

year of the renewed dataset were evaluated.

For the renewed radar product, clutter that was present in the old radar product is effectively removed, showing the efficient application of the fuzzy logic clutter algorithm on Belgian and Dutch volumetric radar data and of the Gabella clutter filter on 2D Cartesian data per radar. Applying additional algorithms, such as a satellite cloud (type) mask and a static clutter mask (Saltikoff et al., 2019; Overeem et al., 2023a), do not seem to be necessary.





The influence of beam blockage due to obstacles and orography is also apparent in maps of annual accumulations, ex-
ceedance probabilities, and metrics at gauge locations. A beam-blockage correction algorithm (Bech et al., 2003; Krajewski
et al., 2006; Lang et al., 2009; Zhang et al., 2013; Cremonini et al., 2016) will likely lead to further improvements of the
product.

Evaluation against independent daily rain gauge accumulations generally reveals a clear improvement for the renewed radar
product in terms of relative bias over the Netherlands (10 percentage points less underestimation), Pearson correlation coeffi-
cient and coefficient of variation. Improvements are also found far away from gauges that are used in the adjustment, likely
pointing to the added value of the VPR correction, rain-induced attenuation correction, quality-based compositing, and the use
of more low-level data, as well as to the contribution of data from an additional German radar. At the same time, the need for
incorporating rain gauge accumulations outside the Netherlands is clear.

Generally, extremes are better captured by the renewed radar product over the Netherlands: underestimations are lower and
Pearson correlation coefficient values are higher. Extremes are very important for many of the applications that use radar-based
QPE, so this is an especially encouraging result. Precipitation underestimation is stronger in the winter period than in the
summer for both products. Quality improvement in the renewed radar product with respect to the old radar product is strongest
in the winter period. This is likely due to the VPR correction and the fact that the quality-based merging algorithm results in
using more low-level data.

The importance of the real-time radar precipitation product expands beyond (near) real-time applications, since it forms the
basis for the early and final reanalysis products after reprocessing. These two products have a longer latency of ∼1 day and
a few weeks, respectively, but their accuracy is higher, since they are enriched with daily manual rain gauge accumulations.
Archives of real-time radar precipitation products can also be useful for training and evaluating (machine learning) nowcasting
algorithms. The KNMI gauge-adjusted real-time precipitation product is updated every 5 min and is the input for the pySTEPS
deterministic nowcasting product (KNMI Radar Team, 2024) and will be used for an upcoming seamless ensemble forecast-
ing product. Hence, any improvements in this precipitation product directly lead to improved nowcasts, forecasts and early
warnings.

## 7.2   Product outlook

Continuous efforts are needed to further improve the real-time radar precipitation product. The KNMI Radar Team consists
of both software developers and researchers, and has adopted the Agile way of working. This allows for fast deployment of
improvements to algorithms and addition of new data sources. This makes this real-time QPE product a living dataset. Research
carried out on product improvement and evaluation is done in a shared environment, which greatly facilitates collaboration
within the team. End users are requested to provide us with feedback on the quality of our product for their specific use case
(radar@knmi.nl).

The evaluation of the renewed radar product is assumed to be representative for the current radar product, although already
some changes or improvements have been implemented in the renewed radar product after the end of the evaluation period (31
January 2024):





– Data from the German radar in Neuheilenbach are added since 9 September 2024, providing better coverage (Fig. 1a).

– Latency of the product has been reduced from on average ∼ 7 min to ∼ 2 min since 9 September 2024.

– The forecasted freezing-level height that is employed for the attenuation correction is taken from ECMWF instead of HARMONIE as of 9 September 2024.

– For the Belgian radar in Jabbeke and the German radars, data from (almost) twice as many elevation scans are added as of 9 September 2024 and a few more for all Belgian and German radars as of 5 November 2024.

– For the adjustment, a dataset of KNMI automatic rain gauge accumulations is employed with a much shorter latency of 5–10 min as of 9 October 2024. This results in a 60 min spatial adjustment factor field being computed every 5 min with a latency of 5–10 min. Hence, the adjustment factor field is much more representative for the current 5 min time interval to which it is applied. An estimate of the resulting improvement is provided in Sect. 5.

– The beam blockage in easterly direction from the Dutch radar in Herwijnen is partly corrected for by assigning a low(er) 
weight to the data from the blocked sector in the two lowest elevation scans since 5 November 2024. Data from other scans and radars will therefore dominate in the final product, hence greatly reducing the effect of beam blockage.

– Rain gauge accumulations from ∼ 150 water authority rain gauges are subjected to quality control (Van Andel, 2021) and used in the computation of the spatial adjustment factor field since 18 November 2024. Their distribution over the land surface of the Netherlands is irregular, although coverage is expected to increase. Currently, these gauges receive a low 
weight in computing the adjustment factor field to prevent overwhelming KNMI's 32 automatic gauges. The weighting and added value of the water authority gauges is still under investigation.

– Improved drizzle detection by avoiding rounding between processing steps, implemented 24 February 2025.

– We have implemented a cap of ±10 dB on the spatial adjustment factor, since 24 February 2025.

In addition, research is in progress to show the potential for improving the real-time radar precipitation product:

– Use differential phase shift to improve precipitation estimation (Testud et al., 2000; Ryzhkov et al., 2014).

– Assess the potential added value of crowdsourced rain gauge accumulations from personal weather stations for adjustment of radar accumulations. This has already been demonstrated at the European scale for 1 h accumulations from 1 year (Overeem et al., 2024a). These data are potentially available in real time and typically have a much higher network density than government gauges in regions covered by weather radars (Overeem et al., 2024b). This connects to the 
rise of opportunistic precipitation sensing, as stimulated by the COST Action OpenSense (https://opensenseaction.eu/, last access: 23 January 2025).

Future potential developments are:





- Beam-blockage correction employing a digital elevation model.

- Improved radar hardware monitoring including alerting.

- Use rain gauge accumulations from Belgian and German (hydro)meteorological services for the adjustment. This is expected to give a major improvement outside the Netherlands, but requires low latency of gauge observations.

- Reduced latency of KNMI's automatic rain gauge data enabling the computation of a true spatial adjustment factor field that incorporates the last 5 min radar precipitation accumulations.

- Employ hydrometeor classification (Al-Sakka et al., 2013) to further improve precipitation retrieval by conversion to a
liquid water content (Smith, 1984; Bukovčić et al., 2018).

- Use $Z - R$ relations specific for the precipitation types (stratiform, convective, undefined; as defined in the VPR correction algorithm).

- Improve the quality index that is contained in the product. This quality index field ranges from 0–1 and serves as a proxy for uncertainty in precipitation estimates. Optimizing parameter settings and relationships for the computation
of the quality index may yield better estimates of precipitation and its uncertainty. This especially concerns the gauge adjustment, that currently results in quality indices nearing one in the product, which seems not realistic.

- Evaluate other gauge-adjustment methods (Barton et al., 2019; Goudenhoofdt and Delobbe, 2009; Winterrath et al., 2012; McKee and Binns, 2015; De Baar et al., 2023; Nielsen et al., 2024; DWD, 2025).



## Appendix A: Computation of quality index

Quality of radar data is estimated for each radar voxel from individual scans. This quality information is applied in three processing steps:

- Construction of Cartesian maps of surface reflectivity from all scans of a given radar.

- Quality-based compositing when combining these maps from all radars.

- Quality-based weighting of radar-gauge pairs in the gauge-adjustment of radar precipitation accumulations.

The procedure to estimate data quality is based on the premise that each correction algorithm is implicitly also a quality assessment. So every source of error, correction, or uncertainty reduces the quality of radar data. This quality reduction factor $Q_X$ (X is an indicator of the correction algorithm) ranges from 0 (not useful: resulting quality is 0) to 1 (very useful: no quality reduction). $Q_X$ is computed for the following correction algorithms:

- Clutter identification by Clutter Correction (CCOR) thresholding (based on the amount of Doppler clutter filtering),
the fuzzy logic algorithm, and the Gabella clutter filter. If any of these algorithms label a voxel or pixel as clutter, the corresponding quality is set to 0. If a voxel is not labeled as clutter, the amount of reflectivity correction by the Doppler clutter filter ($C$ (dB)) leads to the following quality reduction:

$$Q_C = \exp\left(-\ln(2)\left(\frac{|C|}{C_0}\right)^2\right),\tag{A1}$$

with $C_0$ a constant describing the amount of correction associated with halving the quality, here taken as $C_0 = 3$ dB.

- Quality reduction in case of higher probability on clutter (F). The fuzzy logic algorithm generates a score $F$ with a probability that the echo in a given voxel is meteorological (i.e., the weighted average of the degree of membership to the meteorological target class). The fuzzy logic algorithm removes voxels with a score lower than 0.6 (Dutch radars) or 0.3 (Belgian radars). This score is employed to compute a quality reduction for the Belgian and Dutch radar data in case the voxel has not been identified as clutter:

$$Q_F = S\left(F, F_5, F_{95}\right),\tag{A2}$$

where the constants $F_5$ and $F_{95}$ show at which point the quality is 0.05 and 0.95, respectively. Here, $F_5 = 0.8$ and $F_{95} = 0.95$. The function $S$ is a sigmoid function given by:

$$S\left(x, x_5, x_{95}\right) = \frac{1}{1 + \exp\left(\frac{2\log\left(\frac{0.95}{0.05}\right)\left(x - \frac{x_5 + x_{95}}{2}\right)}{x_5 - x_{95}}\right)}.\tag{A3}$$

- Quality reduction by the amount of correction for rain-induced path attenuation that has been applied (A). The uncertainty
as a result of the correction for attenuation is proportional to the amount of correction $A$ (Overeem et al., 2021). This





quality reduction ($Q_\mathrm{A}$) is computed by:

$$Q_\mathrm{A} = \exp\left(-\ln(2)\left(\frac{|A|}{A_0}\right)^2\right),\tag{A4}$$

with $A_0$ a constant describing the amount of correction associated with halving the quality, here $A_0 = 3$ dB.

- Quality reduction by the amount of correction for the vertical profile of reflectivity (VPR) that has been applied (V). This quality reduction $Q_\mathrm{V}$ has the same form as the quality reduction by the Doppler clutter filter (Equation (A1)) and attenuation correction (Equation (A4)). Again halving the quality corresponds to a 3 dB correction.

- Quality reduction by uncertainty in the derived VPR (U). The estimation of the VPR comes with an uncertainty estimate. This uncertainty also results in a quality reduction $Q_\mathrm{U}$ that is computed in the same manner as in Equations (A1) and (A4)), where halving the quality again corresponds to 3 dB uncertainty.

- Quality reduction as a result of the height of the sampling volume above the Earth's surface (H). The higher the sampling volume, the more can happen with the precipitation before it reaches the Earth's surface. This is already partly taken into account by the VPR correction for stratiform and undefined precipitation, but uncertainty will increase with height due to greater differences between precipitation aloft and on the ground (both hydrometeor type and size distributions). In addition, despite the application of algorithms to remove clutter, residual ground clutter could still be present. Hence, the quality close to the Earth's surface is also considered low, but rising steeply in the lower 500 m of the atmosphere. The resulting quality reduction $Q_\mathrm{H}$ as a function of height $h$ is computed by:

$$Q_\mathrm{H} = \frac{S\left(h, 0, h_\mathrm{l}\right) - 0.05}{0.95} S\left(h, h_\mathrm{h}, h_\mathrm{m}\right),\tag{A5}$$

where $0 < h_\mathrm{l} < h_\mathrm{m} < h_\mathrm{h}$. So the quality first increases from the ground to $h_\mathrm{l} = 0.5$ km, followed by a stable high quality ($\sim 0.95$) up to $h_\mathrm{m} = 1.0$ km, after which it decreases again to a value of approximately 0.05 at $h_\mathrm{h} = 4.0$ km. Again, the function $S()$ is the Sigmoid function (Equation (A3)).

- Quality reduction as a result of the distance to the radar (R). The further away from the radar, the larger the sampling volume over which precipitation is averaged (beam broadening). The resulting quality reduction ($Q_\mathrm{Rm}$) as a function of distance $r$ (km) to the radar is assumed to be linear:

$$Q_\mathrm{Rm} = \begin{cases} 1 - \frac{r}{r_\mathrm{m}} & \text{if} \quad r < r_\mathrm{m} \\ 0 & \text{if} \quad r \geq r_\mathrm{m}, \end{cases}\tag{A6}$$

with $r_\mathrm{m}$ the distance (km) at which the quality is reduced to 0. Here, $r_\mathrm{m} = 500$ km. To avoid sharp discontinuities at the maximum range of a radar scan, quality is reduced ($Q_\mathrm{Re}$) in the last $r_\mathrm{e}$ km of the scan:

$$Q_\mathrm{Re} = \begin{cases} 1 & \text{if} \quad r \leq r_\mathrm{max} - r_\mathrm{e} \\ 1 - \left(\frac{r - r_\mathrm{max}}{r_\mathrm{e}} + 1\right)^2 & \text{if} \quad r > r_\mathrm{max} - r_\mathrm{e}, \end{cases}\tag{A7}$$



with $r_\text{max}$ the maximum possible distance to the radar for the chosen scan. This reduction is applied to the last $r_\text{e} = 50\,\text{km}$ of the scan. The total quality reduction ($Q_\text{R}$) as a consequence of the distance to the radar is then:

$$Q_\text{R} = Q_\text{Rm} \cdot Q_\text{Re}. \tag{A8}$$

The total quality index $Q_\text{T}$ is the product of all quality reduction factors described above:

$$Q_\text{T} = Q_\text{C} \cdot Q_\text{F} \cdot Q_\text{A} \cdot Q_\text{V} \cdot Q_\text{U} \cdot Q_\text{H} \cdot Q_\text{R}. \tag{A9}$$

## Appendix B: Computation and application of spatial adjustment factor field

This appendix follows the procedure as described in Overeem et al. (2023a), but with some modifications regarding the settings.

The spatial adjustment factor field $F_\text{adj}$ is calculated as follows for each radar pixel $i$ with position $(x_i, y_i)$:

$$F_\text{adj}(x_i, y_i) = \begin{cases} \frac{S_\text{w,r}}{S_\text{w,g}} & \text{if} \quad S_\text{w,r} > T \bigvee S_\text{w,g} > T, \\ \frac{T}{S_\text{w,g}} & \text{if} \quad S_\text{w,r} \leq T \bigvee S_\text{w,g} > T, \\ \frac{S_\text{w,r}}{T} & \text{if} \quad S_\text{w,r} > T \bigvee S_\text{w,g} \leq T, \\ 1 & \text{if} \quad S_\text{w,r} \leq T \bigvee S_\text{w,g} \leq T, \end{cases} \tag{B1}$$

with $T$ a threshold value of $0.25\,\text{mm}$. The distance-weighted sum of $60\,\text{min}$ precipitation accumulations at gauge locations is denoted by $S_\text{w,X}$, with X an indicator being g (gauge) or r (radar):

$$S_\text{w,X} = \sum_{n=1}^{N_\text{P}} w_n R_\text{X,n}, \tag{B2}$$

which is calculated over $N_\text{P}$ radar–gauge pairs, with the $60\,\text{min}$ precipitation accumulation for the gauge at location $n$ denoted by $R_\text{g,n}$ and the corresponding accumulation for the unadjusted radar data denoted by $R_\text{r,n}$.

In case of low precipitation accumulations, the values for $S_\text{w,X}$ may be very sensitive to errors in both radar and gauge estimates, leading to very uncertain adjustment factors. Especially in cases where radar accumulations are significant at other locations than at the gauges, the absolute effect on the resulting precipitation field may be large. For example, in case of

underestimation by the radar, this may lead to unrealistic extreme accumulations in the final product. To prevent this, the value of $S_\text{w,X}$ is set equal to $T$ when it becomes smaller than $T$ (see Eq. B1).

The weighting function $w_n$ is a function of the distance of a pixel to a gauge location $n$ and of the quality of radar and rain gauge data:

$$w_n = \frac{G_\text{w}(n, r_\text{s}) + v \cdot G_\text{w}(n, r_\text{l})}{1 + v} Q_\text{r,n} Q_\text{g,n}, \tag{B3}$$

where $G_\text{w}(n, r_\text{d})$ is a Gaussian function:

$$G_\text{w}(n, r_\text{d}) = \begin{cases} \frac{\exp\left(-4 \frac{\delta_{i,n}^2}{r_\text{d}^2}\right) - \exp(-4)}{1 - \exp(-4)} & \text{if} \quad \delta_{i,n}^2 \leq r_\text{d}, \\ 0 & \text{if} \quad \delta_{i,n}^2 > r_\text{d}, \end{cases} \tag{B4}$$





in which $\delta_{i,n} = \sqrt{(x_i - x_n)^2 + (y_i - y_n)^2}$ is the distance between the gauge and the radar pixel, and $(x_n, y_n)$ is the position of gauge $n$. Note that the quality of both gauge data ($Q_{g,n}$) and radar data ($Q_{r,n}$, see Appendix A) is taken into account. In the current implementation, the quality of rain gauge data is set to a fixed value of 0.9 (assumed quality due to measurement and representativeness errors). If gauge quality differs across the network, this algorithm is hence able to take that into account. The weighting consists of a short-range and a long-range component, each having its own Gaussian function in Equation (B3). The Gaussian function decreases to 0 when a pixel is located a range of at least $r_d$ km from the selected gauge location $n$. The long-range component contributes up to a range of 500 km ($r_l$) and is meant for adjusting when the distance to nearest gauges is long, typically further away from the Netherlands. The short-range component provides a much more local adjustment up to a range of $r_s$, which is based on the range of a seasonally-dependent isotropic spherical variogram model of rainfall accumulations corresponding to a 1 h duration (Van de Beek et al., 2012). The value of $v$ controls the contribution of the long-range component with respect to the short-range component, and is set to $v = 0.1$ in the current implementation.

The real-time radar precipitation product of 5 min accumulation ($R_{r,adj}$; mm) at a radar pixel $i$ with position $(x_i, y_i)$ is obtained as follows:

$$R_{r,adj}(x_i, y_i) = \frac{R_r(x_i, y_i)}{10^{F_{adj}(x_i, y_i)/10}}, \tag{B5}$$

with $R_r$ the 5 min unadjusted accumulation (mm), and $F_{adj}$ the spatial adjustment factor field (see Eq. B1).

A quality index field ($Q$) is also computed for each pixel. Each rain gauge measurement that is added to the product should increase the quality, and the resulting quality index should be between 0 and 1. This leads to the following expression for the product quality:

$$Q_i = Q_{r,i} \left( 1 - \prod_{n=1}^{N_p} (1 - w_n Q_{g,n}) \right), \tag{B6}$$

with $Q_{r,i}$ the radar quality at pixel $i$ (Equation (2)) and $Q_{g,n}$ the quality of the observation from rain gauge $n$.



*Supplement.* The supplemental video (Movie S1) shows an animation of 5 minute precipitation accumulations from the renewed product during summer storm Poly on 5 July 2023 from 03:55 UTC – 08:00 UTC over the Netherlands. The supplement related to this article is available online at https://doi.org/10.5194/essd-0-1-2025-supplement.

*Author contributions.* Author contributions are captured following the CRediT system. Conceptualization: AO and HL. Data curation: AO. Formal analysis: AO. Funding acquisition: HL. Investigation: AO. Methodology: AO and HL. Project administration: AO and HL. Software: AO, BA, HL and MV. Supervision: AO and HL. Validation: AO. Visualization: AO. Writing – original draft preparation: AO and HL. Writing – review and editing: AO, BA, HL and MV.

*Competing interests.* The contact author has declared that none of the authors has any competing interests.

*Disclaimer.* Publisher's note: Copernicus Publications remains neutral with regard to jurisdictional claims in published maps and institutional affiliations.

*Acknowledgements.* We thank Xueli Wang and Bastiaan Anker (KNMI) for the operationalization of the real-time radar product processing. We thank Tim Vlemmix (KNMI) for providing comments on a draft of this manuscript. We are grateful for the E-OBS dataset (Cornes et al., 2018) from the EU-FP6 project Uncertainties in Ensembles of Regional ReAnalyses (https://www.uerra.eu, last access: 23 January 2025) and
from the Copernicus Climate Change Service, for the data providers in the ECA&D project (https://www.ecad.eu, last access: 23 January 2025) and for the Flanders Environmental Agency (VMM), Royal Meteorological Institute of Belgium (RMI) and the German Weather Service (DWD) for providing the radar data. Finally, we thank DWD for their publicly available rain gauge data. All figures containing maps were made with the Python package Cartopy (Met Office, 2022).

*Financial support.* We gratefully acknowledge funding from "Slim Water Management" program administered by Rijkswaterstaat on behalf
of the Dutch Ministry of Infrastructure and Water Management, from the Foundation for Applied Water Research STOWA, and from "het Waterschapshuis," representing the 21 Dutch water boards, for the project "Onderzoek neerslagmetingen." This funded the research to develop and test the radar processing algorithms.





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



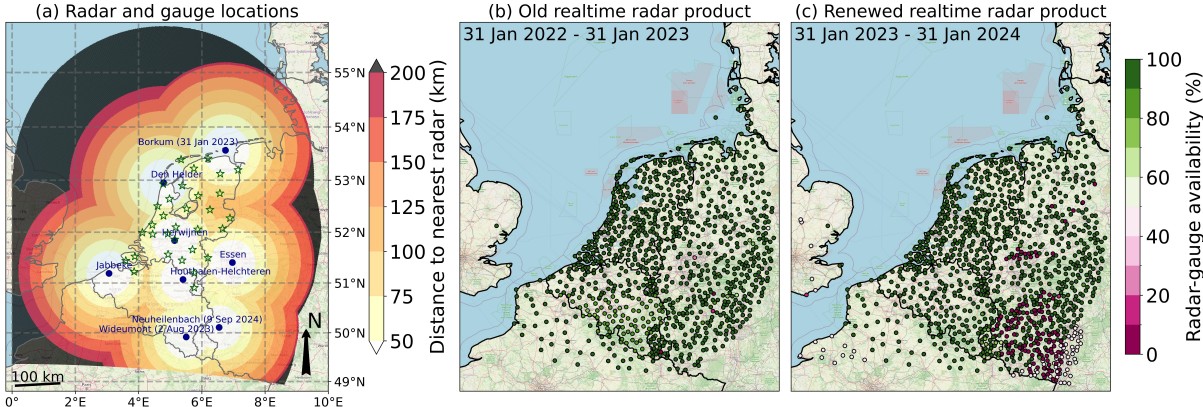

**Figure 1. (a)** Map with radar (dots) and automatic rain gauge (stars) locations. Background color indicates the distance to the nearest radar assuming full availability of radar data (note that some radars, or scans, only contributed part of the period, and the Neuheilenbach radar did not contribute to the datasets studied in this paper) and taking into account their maximum range. **(b,c)** Map with combined radar–gauge availability for the daily accumulations from the old (31 January 2022 – 31 January 2023) and the renewed (31 January 2023 – 31 January 2024) real-time QPE product. Map data © OpenStreetMap contributors 2025. Distributed under the Open Data Commons Open Database License (ODbL) v1.0.





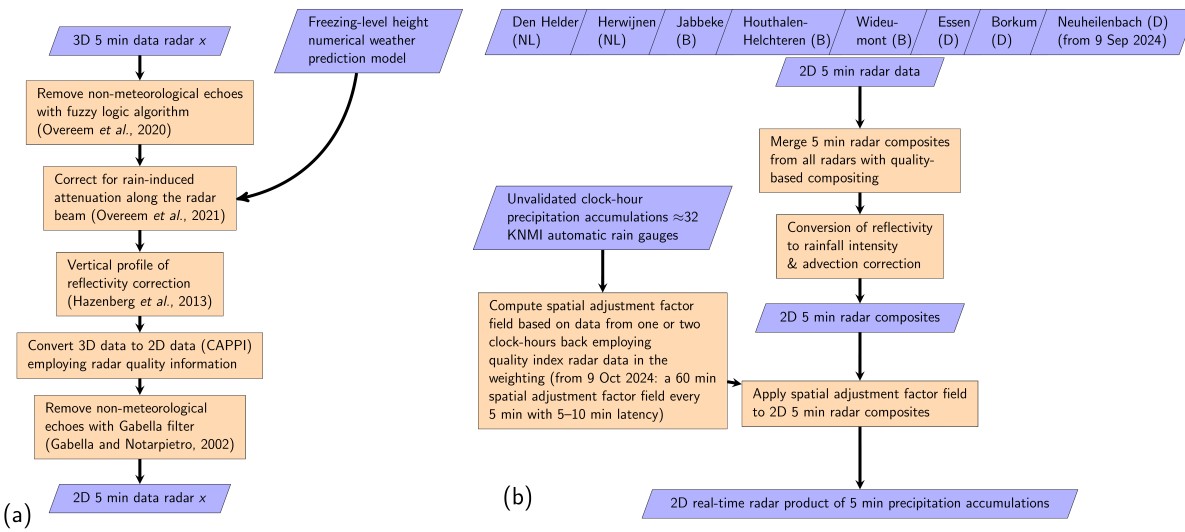

(a) (b)

**Figure 2.** Flowcharts of the real-time QPE production chain: **(a)** for single radar processing; **(b)** for the combined radar and rain gauge data.



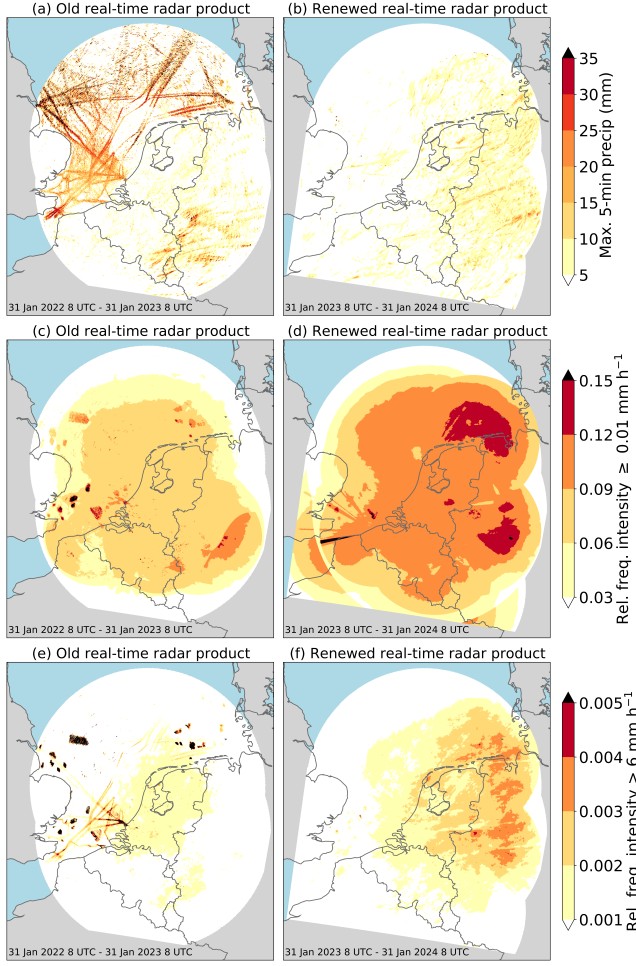

**Figure 3. (a,b)** Maps of maximum 5 min precipitation accumulation and **(c–f)** maps of the relative frequency of 5 min precipitation ≥0.01 mm
h$^{-1}$ and >6 mm h$^{-1}$. For the old (left panels) and renewed (right panels) radar product. No data availability criterion has been applied. Maps
made with Natural Earth. Free vector and raster map data © https://naturalearthdata.com (last access: 23 January 2025).

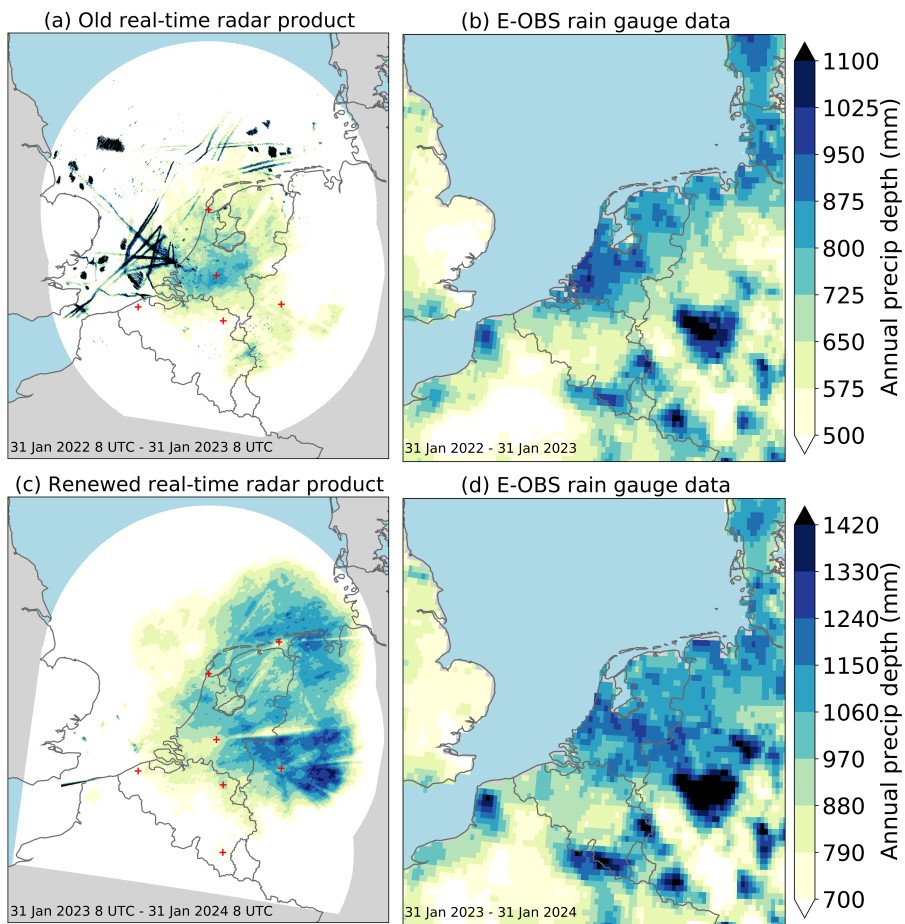

**Figure 4.** Annual precipitation accumulations for **(a)** the old radar product ($\sim$1 km$^2$) and for **(b)** interpolated rain gauge observations (E-OBS version 30.0e; release September 2024; $0.1° \times 0.1°$). **(c,d)** show the accumulations for the renewed radar product and E-OBS. Maps made with Natural Earth. Free vector and raster map data © https://naturalearthdata.com (last access: 23 January 2025).



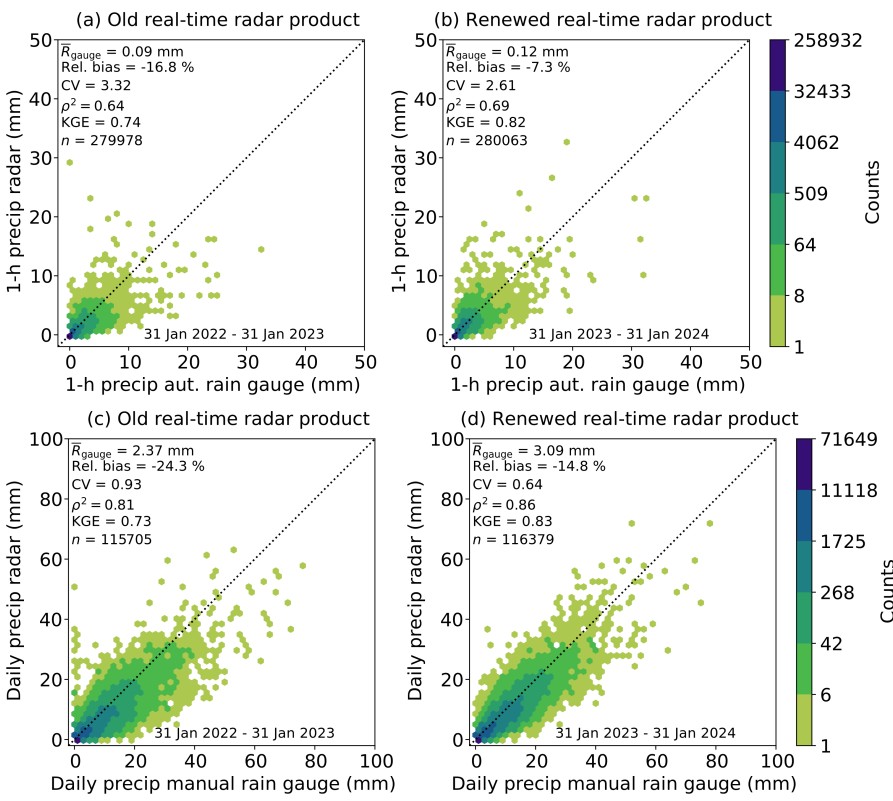

**Figure 5.** Scatter density plots of **(a,b)** 1 h and **(c,d)** daily radar precipitation accumulations against, respectively, independent automatic and manual KNMI rain gauge accumulations from the Netherlands for the old (left panels) and renewed (right panels) radar product.

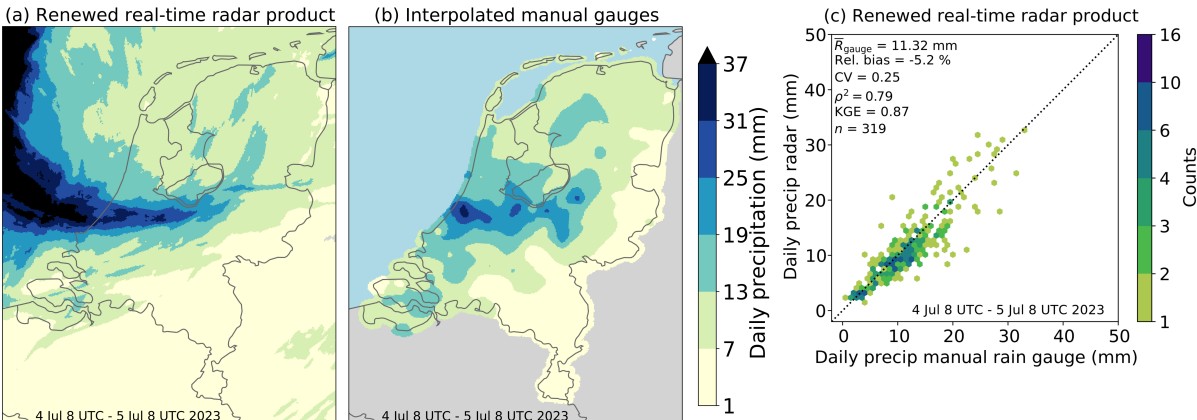

**Figure 6.** Daily precipitation accumulations caused by summer storm Poly for **(a)** the renewed radar product and for **(b)** independent interpolated daily manual rain gauge observations for the Netherlands. **(c)** Scatter density plot of daily radar precipitation accumulations for the renewed radar product against independent rain gauge accumulations from KNMI's manual network from the Netherlands. The 5 min precipitation accumulations from 5 July 2023 03:55 UTC – 08:00 UTC are presented as the Supplement (Movie S1). Maps made with Natural Earth. Free vector and raster map data © https://naturalearthdata.com (last access: 23 January 2025).

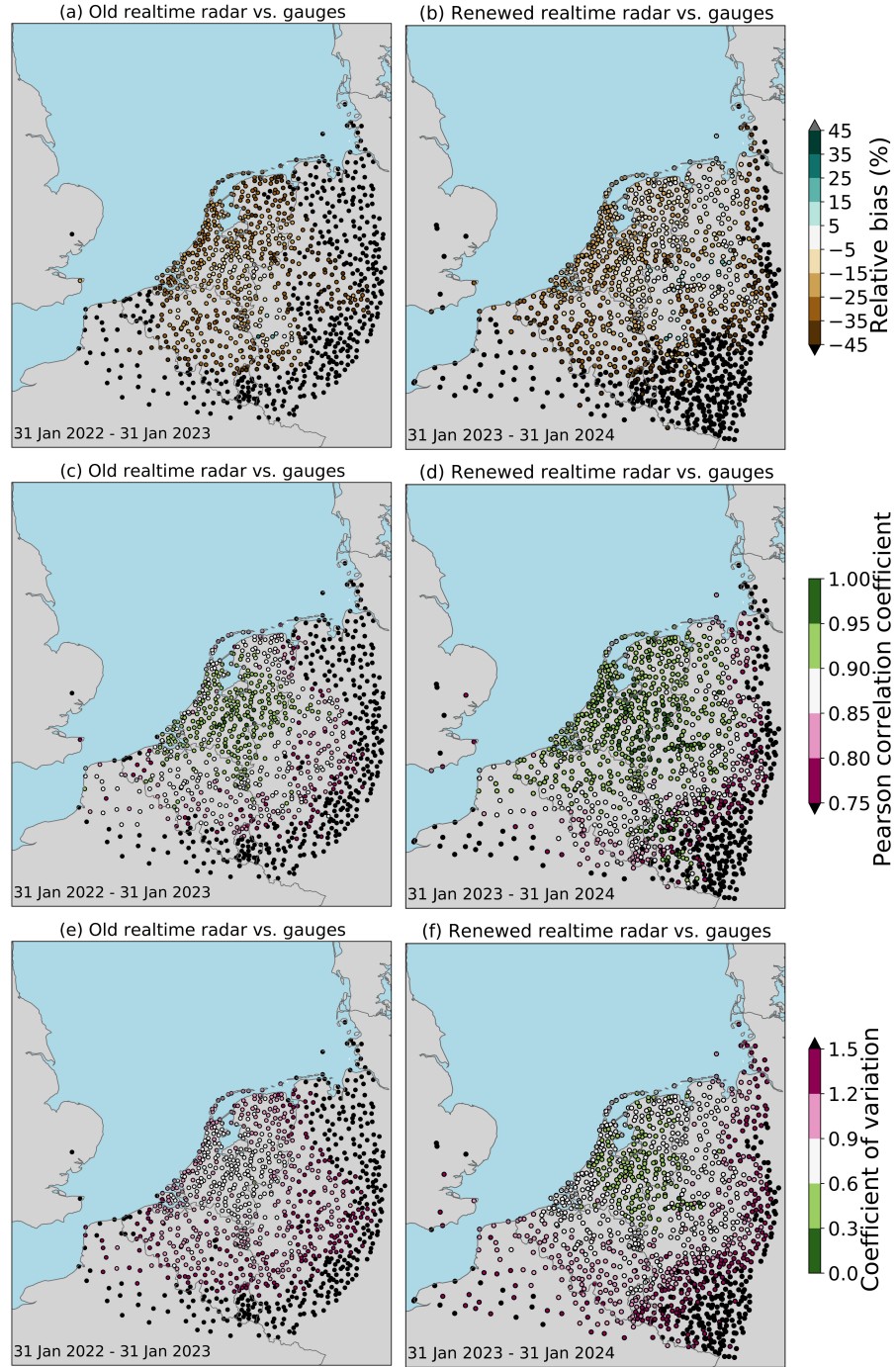

**Figure 7.** Independent spatial evaluation of daily radar precipitation accumulations against the daily rain gauge precipitation accumulations for **(a, c, e)** the old radar product (1008 gauge locations) and for **(b, d, f)** the renewed radar product (1210 gauge locations). Note that the joint radar-gauge availability is lower for some locations (Fig. 1a–b). Maps made with Natural Earth. Free vector and raster map data © https://naturalearthdata.com (last access: 23 January 2025).

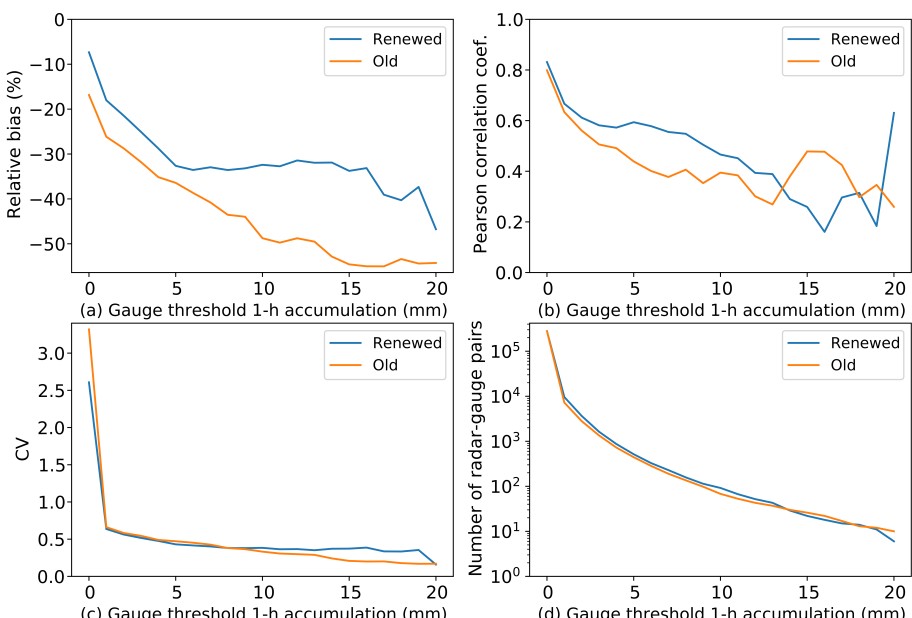

**Figure 8.** Independent evaluation of radar 1 h precipitation accumulations against the 32 KNMI automatic rain gauge accumulations from the Netherlands for the old (orange lines) and renewed (blue lines) radar product for gauge threshold values ranging from 0 mm (no thresholding) to 20 mm.

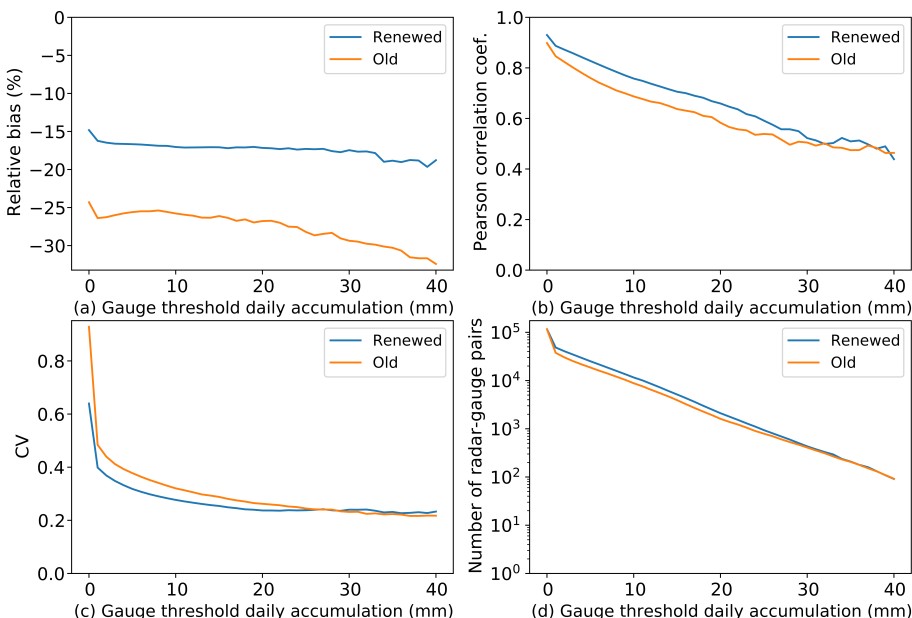

**Figure 9.** Independent evaluation of radar daily precipitation accumulations against the 317–319 KNMI manual rain gauge accumulations from the Netherlands for the old (orange lines) and renewed (blue lines) radar product for gauge threshold values ranging from 0 mm (no thresholding) to 40 mm.

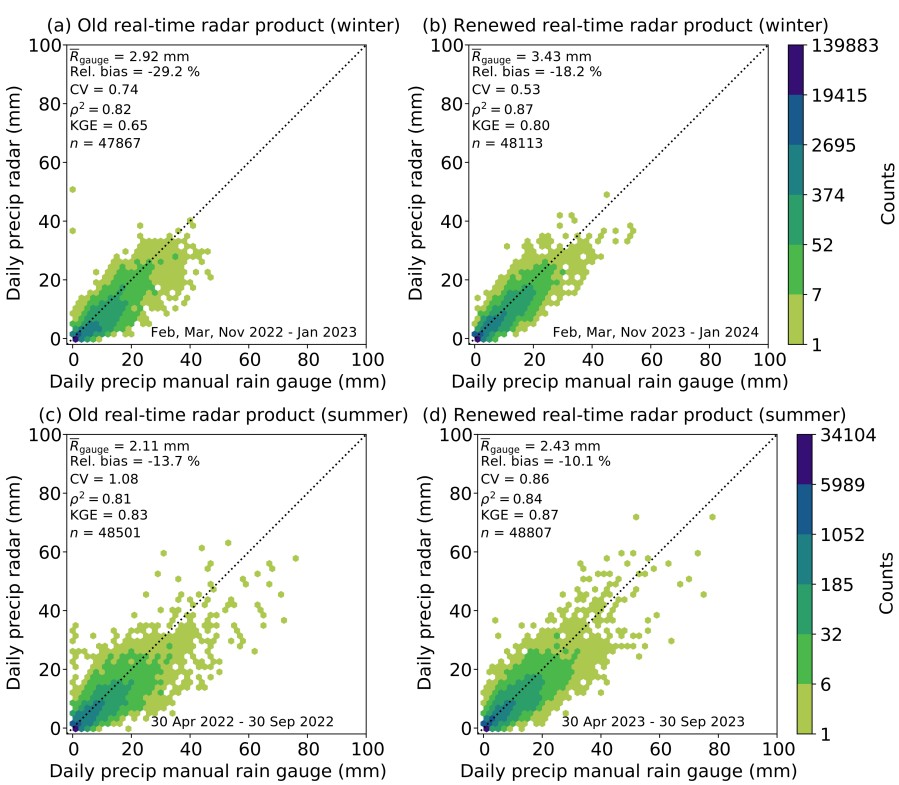

**Figure 10.** Scatter density plots of daily radar precipitation accumulations against independent manual KNMI rain gauge accumulations from the Netherlands for the old (left panels) and the renewed (right panels) radar product for **(a,b)** the winter period and for **(c,d)** the summer period.