# Peer review of "The Dutch real-time gauge-adjusted radar precipitation product"

_Earth System Science Data, 2025_

## Referee Comment (RC1)

Review of the manuscript "The Dutch real-time gauge-adjusted radar precipitation product" submitted to ESSD.

This manuscript describes in detail the new precipitation product over the Netherlands and surrounding areas. It compares its performance to the previous version of the product by analyzing statistics over a full year (the last year of the former product, and the first year of the renewed product).

The manuscript is well written and mostly well explained. The dataset it describes is of great utility for many applications.

I have a few remarks about the radar data quality and the radar rainfall retrieval, and some minor comments listed below.

I think it should be published in ESSD after addressing these points.

Major points:

1) Calibration of the radars is only mentioned in the discussion. Does this mean there is no prior calibration performed? Given that the radars used belong to multiple agencies, it is not clear if and how the calibration is performed.

2) I wonder why the rainfall retrieval does not make use of polarimetry. The polarimetry is used for the clutter identification, the attenuation correction but not for improving the actual retrievals. Z-R relationships have been shown to perform much worse than polarimetric retrievals (Specific differential phase or more recently specific attenuation, or even hybrid retrievals). See for example Chen et al .2021.

Chen, J., S. Trömel, A. Ryzhkov, and C. Simmer, 2021: Assessing the Benefits of Specific Attenuation for Quantitative Precipitation Estimation with a C-Band Radar Network. *J. Hydrometeor.*, **22**, 2617–2631, https://doi.org/10.1175/JHM-D-20-0299.1.

And why the choice of the Marshall-Palmer relationship? Is this relation the most adequate to this region?

3) Concerning the topic of beam blockage, why not include a quality index associated with it and reduce the weight in the pixels affected by it? And also why not mitigate the problem with polarimetry? Using phase based retrievals could improve the product where partial beam blockage is a problem. Seems like the tools were all there.

4) The gauge adjustments and the adjustment that is only performed the next hour, this was not clear to me. Maybe I misunderstood, but it is not logic to me to adjust radar rainfall with gauge measurements from another time interval.

Minor comments:

Lines 84-85: I think the part "except for the renewed radar product....." is related to the previous sentence and makes no sense here after mentioning the cause of the lower availability.

Line 153: Why only mention the weights for this case and not for the Dutch radars?

Lines 170-175: There seems to be a lot of repetition in this paragraph. Please rewrite in a more clear way.

Lines 175-176: Check this sentence I think it should be "NO attenuation correction is applied to voxels above…"

Line 182: If you are showing the relationship for Kv, why not show the equivalent relationship for Kh?

Lines 611-612: I am a bit confused here. Should it maybe be "higher than 6" instead of "fewer than 6"?

Section 4, when the performance scores are presented: Could the authors justify the reason these scores were chosen instead of more commonly used scores like RMSE, MAE?

Lines 285-286: "Clutter for the land surface seems less of an issue for the old radar product, and even less so for the renewed radar product" I don't think this statement is accurate. Over the NL the problem seems worse.

Line 292: Suggest changing "will be" to "could be".

Line 303: Could the authors try to explain such a severe underestimation to the east of the Essen radar in figure 4 a)? There is absolutely no rain (blank in the figure) but RG measures values > 1000 mm? And this is so close to the Essen radar, what is happening here?

Lines 324-325: This sentence is not very clear, please rewrite: "The underestimation for true precipitation events in the old radar product is expected to be larger, because of compensation by clutter leading to overall less underestimation"

Section 4.5 To better understand this section it would be nice to include a discussion on the spatial variability of the indexes and the quality.

Line 349: "German Weather Service has already corrected their radar data for attenuation, so it has been corrected for attenuation twice"

This doesn't make any sense, why would you correct twice for attenuation? Obviously this will introduce errors.

Line 350: Suggest changing "As for" to "Similar to"

Section 4.6: What is the conclusion from these results? Comment on the performances for each thresholds for both products.

Line 395: Suggest changing to "may be the best option".

Line 411: applied instead of applies.

Line 476: What is "Agile"?

Figure 1: The colored points in b and c are very difficult to see, this figure needs to be enlarged.

Figure 4: Mention what the red crosses are in the legend.

---

## Author Comment (AC1)

**RC1**: 'Comment on essd-2025-160', Anonymous Referee #1, 10 May 2025

Review of the manuscript "The Dutch real-time gauge-adjusted radar precipitation product" submitted to ESSD.

This manuscript describes in detail the new precipitation product over the Netherlands and surrounding areas. It compares its performance to the previous version of the product by analyzing statistics over a full year (the last year of the former product, and the first year of the renewed product).

The manuscript is well written and mostly well explained. The dataset it describes is of great utility for many applications.

I have a few remarks about the radar data quality and the radar rainfall retrieval, and some minor comments listed below.

I think it should be published in ESSD after addressing these points.

**We thank the reviewer for the constructive assessment of our manuscript.**

Major points:

1) Calibration of the radars is only mentioned in the discussion. Does this mean there is no prior calibration performed? Given that the radars used belong to multiple agencies, it is not clear if and how the calibration is performed.

**Calibration of radars is the responsibility of the individual institutes that own the radars. Calibration of the Dutch radars is described in Beekhuis and Mathijssen (2018). For the Dutch radars half-yearly servicing includes single point calibration of the (receive) path, calibrate receiver channel separation, calibrate transmission power and ZDR calibration. More work is needed to increase the number of monitoring tools and their operational use. For the German radars, details on calibration and monitoring are provided by Frech et al. (2017): it includes a 90-degree elevation (bird-bath) scan every 5 min and a clutter target scan every hour. "The calibration scan performs a one-point calibration using the built-in internal test signal generator". The radars are maintained every 9 months and then four calibrations are carried out. For the Belgian radars in Jabbeke and Wideumont, we contacted the Royal Meteorological Institute of Belgium. They confirm that solar monitoring and bird-bath scan are performed (pers. com.).**

**We added to the radar data subsection: "All radars that are used in this product are calibrated during regular maintenance (1-2 times per year). Monitoring tools are in place at each institute, which can assist to detect drifts in calibration so that action can be taken if this is the case. It is therefore assumed that the calibration of the radars is accurate. For more information on calibration and monitoring, see**

**Beekhuis and Mathijssen (2018) for the Dutch radars, and Frech et al. (2017) for the German radars."**

2) I wonder why the rainfall retrieval does not make use of polarimetry. The polarimetry is used for the clutter identification, the attenuation correction but not for improving the actual retrievals. Z-R relationships have been shown to perform much worse than polarimetric retrievals (Specific differential phase or more recently specific attenuation, or even hybrid retrievals). See for example Chen et al .2021.

Chen, J., S. Trömel, A. Ryzhkov, and C. Simmer, 2021: Assessing the Benefits of Specific Attenuation for Quantitative Precipitation Estimation with a C-Band Radar Network. J. Hydrometeor., 22, 2617–2631, https://doi.org/10.1175/JHM-D-20-0299.1.

**We acknowledge the potential of polarimetry for rainfall retrieval and have already performed research on the use of a hybrid *Z*-Kdp rainfall estimation algorithm for Dutch and Belgian radars. Results were not entirely conclusive. Hence, we decided to start investigating the specific attenuation-based method from Ryzhkov et al. (2014), as is already noted in our manuscript in the product outlook section: "Use differential phase shift to improve precipitation estimation in case of rain (Testud et al., 2000, Ryzhkov et al., 2014).". For frozen precipitation and the transition between frozen and liquid precipitation this requires more attention before we can use it in our operational product.**

And why the choice of the Marshall-Palmer relationship? Is this relation the most adequate to this region?

**The employed Marshall-Palmer relationship is known for its representativeness for temperate climates, which includes the Netherlands. It is "considered to be fairly typical" for stratiform rainfall (Battan, 1973), which is the dominant rainfall type in the Netherlands. It has been shown to be representative for the Netherlands (Wessels, 1972; Holleman, 2006). One could consider the use of multiple *Z-R* relationships to obtain a better fit for, e.g., convective rainfall, as is done in Germany having a similar climate. Our current focus is to invest all time related to rainfall retrieval in methods partly based on polarimetry.**

3) Concerning the topic of beam blockage, why not include a quality index associated with it and reduce the weight in the pixels affected by it? And also why not mitigate the problem with polarimetry? Using phase based retrievals could improve the product where partial beam blockage is a problem. Seems like the tools were all there.

**As stated in the product outlook section, we already use a lower quality index to mitigate beam blockage by trees very close to the Herwijnen radar site. But for other azimuths and radars, beam blockage is not corrected for yet. We performed a detailed study and developed a software tool employing a lidar-based digital**

surface model to detect beam blockage for our Dutch radars (Overeem et al., 2023b). This gave a lot of insight but also showed that very close to radar sites beam blockage detection is still challenging. Moreover, for German radars, we are not aware of any publicly available digital surface models with high spatial resolution, apart from those coarser satellite-based digital elevation models that are useful in case of stronger orography. We are in the process of systematically employing a digital surface model for our Dutch radars to determine the fraction of the beam that is blocked under normal propagation conditions. We expect that this will at least lead to corrections for the most severe beam-blockage affected areas for our Dutch radars, through lowering the quality index and hence the weight of that scan/radar in the final product. Polarimetry could partly help to mitigate the problem because differential phase is immune to partial beam blockage in the sense that it is a phase measurement and not a power measurement. Note though, that rainfall retrieval through specific differential phase at far range from a radar may not be possible because of solid precipitation resulting in noisy values. Moreover, the part of the measurement volume that is blocked, will still not be sampled. We added to the product outlook section (italic font): "Beam-blockage correction employing a digital elevation model. *Blocked sectors from the affected elevation scan are assigned a lower weight. Rainfall retrieval through differential phase can also help to further mitigate beam blockage, although it still does not sample the blocked part of the measurement volume.*"

4) The gauge adjustments and the adjustment that is only performed the next hour, this was not clear to me. Maybe I misunderstood, but it is not logic to me to adjust radar rainfall with gauge measurements from another time interval.

**Ideally, the adjustment is performed on radar data that are from the same time interval as the rain gauge data. But this is not possible due to latency of rain gauge data. This was already explained in subsection 3.9. As already stated in the product outlook section, as of 9 October 2024, the latency of rain gauge data is much shorter (5–10 min), making the adjustment factor field much more representative for the current time interval.**

Minor comments:

Lines 84-85: I think the part "except for the renewed radar product....." is related to the previous sentence and makes no sense here after mentioning the cause of the lower availability.

**Thank you for noting this. To clarify, we changed "Lower availability is caused by missing gauge records, except for the renewed radar product for the (south)western and southeastern part of the product domain." to "Lower availability is caused by missing gauge records. Exception is that for the renewed**

radar product the lower availability for the (south)western and southeastern part of the product domain is caused by lower radar data availability."

Line 153: Why only mention the weights for this case and not for the Dutch radars?

**We originally did not add the weights, because these are provided in Overeem et al. (2020). We have now added the weights of the decision variables for the Dutch radars to make our manuscript more self-contained.**

Lines 170-175: There seems to be a lot of repetition in this paragraph. Please rewrite in a more clear way.

**This concerns the paragraph: "For both methods, the wradlib open-source Python library for weather radar data processing (Heistermann et al., 2013; Mühlbauer et al., 2020) is utilized. Since the correction is meant for attenuation caused by liquid precipitation, a voxel is only used to *compute* the attenuation in case its height is below the forecasted freezing-level height from the numerical weather prediction model HARMONIE-AROME. And for the method using Kdp , voxels classified as clutter are not employed to *compute* attenuation (Wradlib, 2020c). Moreover, attenuation correction is only *applied* to voxels that are not classified as clutter according to the fuzzy logic algorithm (for the Belgian and Dutch radar data). Note that attenuation correction is *applied* to voxels above the freezing-level height."**

**We claim that there is no repetition and that most of these details are needed to make clear what is done in the computation of attenuation (highlighted above by *compute*) and in the actual correction for attenuation (highlighted above by *applied*). Hence, this may be interpreted as repetition, but computation and application are two different steps. But we do acknowledge that some details add to the confusion. Hence, we rewrote "Moreover, attenuation correction is only *applied* to voxels that are not classified as clutter according to the fuzzy logic algorithm (for the Belgian and Dutch radar data). Note that attenuation correction is *applied* to voxels above the freezing-level height." as follows: "Attenuation correction is *applied* to all voxels, so also those above the freezing-level height."**

Lines 175-176: Check this sentence I think it should be "NO attenuation correction is applied to voxels above…"

**The reviewer refers to "Note that attenuation correction is applied to voxels above the freezing-level height.". This statement is correct but has been clarified: "Attenuation correction is *applied* to all voxels, so also those above the freezing-level height."**

**Although the attenuation is only computed with data from voxels below the freezing-level height, the signals from voxels that are above the freezing-level**

height have been affected by attenuation at lower heights. Hence, these are also corrected for attenuation.

Line 182: If you are showing the relationship for Kv, why not show the equivalent relationship for Kh?

**We added the relationship between horizontal specific attenuation and specific differential phase, by changing the text as follows (changes in italic font): "*Horizontal specific* attenuation is computed from specific differential phase ($K_{dp}$) (Bringi et al., 1990) assuming a linear relation between the two for the Belgian and Dutch radars *($k_h = 0.081\ K_{dp}$)*."**

Lines 611-612: I am a bit confused here. Should it maybe be "higher than 6" instead of "fewer than 6"?

**The reviewer is referring to lines 211-212. It should be "fewer than 6". We write: "For each central pixel, the number of pixels within a square lattice of 5 × 5 pixels is counted that have a $Z_{h,Q}$ value less than 6 dB lower than the central pixel. The central pixel is classified as clutter if this number of pixels is fewer than 6."**

**When a pixel is less than 6 dB lower than the central pixel, the pixel values are relatively similar. When fewer than 6 surrounding pixels are less than 6 dB lower than the central pixel, at least 19 surrounding pixels are at least 6 dB lower than the central pixel (i.e., very high spatial variability). This points to clutter in the central pixel.**

Section 4, when the performance scores are presented: Could the authors justify the reason these scores were chosen instead of more commonly used scores like RMSE, MAE?

**We added to the beginning of section 4, where the metrics are described:**

**"The chosen metrics are often employed for verification purposes. The coefficient of variation of the residuals (CV) is the standard deviation of the residuals divided by the mean of the reference, here rain gauges. Values for root mean square error (RMSE) will generally increase for regions with on average higher rainfall amounts. The reason to use CV instead of RMSE is that it facilitates comparisons between studies, because it takes the average climatic conditions into account by normalizing with the mean rainfall of the reference. Furthermore, CV is a better measure of the scatter than RMSE because it is not affected by the bias. MAE is a similar metric, but puts less emphasis on outliers than RSME and CV, but it is also affected by the bias. The Pearson correlation coefficient describes the degree of covariation between the estimate and the reference, hence adding more insight into the performance."**

Lines 285-286: "Clutter for the land surface seems less of an issue for the old radar product, and even less so for the renewed radar product" I don't think this statement is accurate. Over the NL the problem seems worse.

**What we want to convey here is that sea clutter is much more extensive than clutter for the land surface, as can be seen in Fig. 3a,c,e and Fig. 4a for the old radar product, showing many locations with sea clutter. This difference is indeed not that clear for the renewed radar product. Hence, we replaced "Clutter for the land surface seems less of an issue for the old radar product, and even less so for the renewed radar product" by: "Clutter for the land surface is less of an issue than sea clutter, which is clearly visible for the old radar product."**

Line 292: Suggest changing "will be" to "could be".

**We changed this accordingly.**

Line 303: Could the authors try to explain such a severe underestimation to the east of the Essen radar in figure 4 a)? There is absolutely no rain (blank in the figure) but RG measures values > 1000 mm? And this is so close to the Essen radar, what is happening here?

**This is related to the fact that for the old radar product, the weight of a given radar is very small up to ~50 km range. Hence, close to the Essen radar, data are mostly from radars that are quite far away (Herwijnen and Houthalen-Helchteren). This leads to underestimates because of VPR and attenuation effects (that are not corrected for in the old product). Note that blank areas in this figure indicate values below 500 mm (and not no rain), which we agree with the reviewer are gross underestimates of the true precipitation. Note that we already mention in the manuscript: "The large area with high precipitation accumulations in Germany, that is missed in the old radar product, is captured by the renewed radar product. This likely shows the added value of the VPR correction, the rain-induced attenuation correction, the contribution of the Borkum radar, and the much higher weight of data from the nearest radar in the compositing."**

**To emphasize the importance of the low weight of a given radar at short range, we changed the text in: "The large area with high precipitation accumulations in Germany, that is missed in the old radar product, is captured by the renewed radar product. This is likely the result of the much higher weight of data from the nearest radar in the compositing, and to a lesser extent caused by the added value of the VPR correction, the rain-induced attenuation correction, and the contribution of the Borkum radar."**

Lines 324-325: This sentence is not very clear, please rewrite: "The underestimation for true precipitation events in the old radar product is expected to be larger, because of compensation by clutter leading to overall less underestimation"

**We rewrote this sentence as follows: "Since clutter leads to overestimation and is more severe for the old radar product, it partly compensates for the mean underestimation. This implies that the underestimation for true precipitation events in the older radar product is actually larger than indicated in Fig. 5."**

**This is then followed by the already existing sentences: "This is confirmed by the many outliers for (near) zero daily gauge accumulations (and a few outliers for 1 h accumulations), that are not visible anymore in the renewed radar product. Hence, the improvement in the bias may be even larger."**

Section 4.5 To better understand this section it would be nice to include a discussion on the spatial variability of the indexes and the quality.

**In our view, the spatial variability of the relative bias metrics is already described quite extensively. We agree that the description for the other metrics, Pearson correlation coefficient and CV, is limited. Therefore, we added this to the text in Section 4.5: "For the old radar product, values for $\rho$ are typically 0.8 or higher for the land surface of the Netherlands, but rapidly decrease outside the Netherlands to values below 0.75. For the renewed radar product, values for $\rho$ are generally 0.9 or higher for the land surface of the Netherlands, and this is even found for regions in Belgium and Germany that are close to the Dutch border. At farther distances from the Dutch border, values decrease to values below 0.75. Similar behavior is found for CV. To conclude, the quality of the renewed radar product is much better for the land surface of the Netherlands and Belgian and German regions close to the Dutch border.".**

Line 349: "German Weather Service has already corrected their radar data for attenuation, so it has been corrected for attenuation twice"

This doesn't make any sense, why would you correct twice for attenuation? Obviously this will introduce errors.

**We thought that the obtained reflectivity data had not undergone attenuation correction. Unfortunately, the metadata did not indicate the application of attenuation correction. This is why we write "This indicates the importance of provenance in radar metadata.". We clarified this by adding: "This was not done on purpose, but only because it was assumed that attenuation correction had not been applied yet. Since its application was not mentioned in the metadata, …"**

Line 350: Suggest changing "As for" to "Similar to"

**We modified this accordingly.**

Section 4.6: What is the conclusion from these results? Comment on the performances for each thresholds for both products.

**We added: "To summarize, for the renewed radar product the underestimation is approximately 5-20 percentage points less than for the old radar product for the full range of 1 h and 24 h threshold values. The values for $\rho$ are (slightly) higher up to the more extreme 1 h or 24 h threshold values for the renewed product. The values for CV are equal (1 h) or higher (24 h) up to the more extreme threshold values and become lower (1 h) or equal (24 h) for the most extreme threshold values for the renewed radar product."**

Line 395: Suggest changing to "may be the best option".

**We modified this accordingly.**

Line 411: applied instead of applies.

**We modified this accordingly.**

Line 476: What is "Agile"?

**This is a way of working together, especially known from software development, by splitting up tasks and by implementing improvements continuously, instead of finishing a project at once. As such, it can help to avoid delays and focus on a minimum viable product. We added a reference to the Agile Alliance website (Agile Alliance, 2025).**

Figure 1: The colored points in b and c are very difficult to see, this figure needs to be enlarged.

**We enlarged the separate figures, so that Figure 1 now occupies two rows.**

Figure 4: Mention what the red crosses are in the legend.

**We added to the figure 4 caption: "The red crosses in (a,c) denote the radar locations."**

**Bibliography**

**Agile Alliance: What is Agile? | Agile 101 | Agile Alliance, https://www.agilealliance.org/agile101/, retrieved May 2025, 2025.**

**Battan, L. J., 1973: Radar observation of the atmosphere. Revised edition. The University of Chicago Press. Q.J.R. Meteorol. Soc., 99: 793-793.**
**https://doi.org/10.1002/qj.49709942229**

Beekhuis, H. and Mathijssen, T.: From pulse to product, Highlights of the upgrade project of the Dutch national weather radar network, in: 10th European Conference on Radar in Meteorology and Hydrology (ERAD 2018) : 1-6 July 2018, Ede-Wageningen, The Netherlands, edited by de Vos, L., Leijnse, H., and Uijlenhoet, R., pp. 960–965, Wageningen University & Research, Wageningen, the Netherlands, https://doi.org/10.18174/454537, 2018.

Frech, M., M. Hagen, and T. Mammen, 2017: Monitoring the Absolute Calibration of a Polarimetric Weather Radar. J. Atmos. Oceanic Technol., 34, 599–615, https://doi.org/10.1175/JTECH-D-16-0076.1.

Holleman, I., 2006: Bias adjustment of radar-based 3-hour precipitation accumulations. Technical report TR-290, KNMI, De Bilt, https://cdn.knmi.nl/knmi/pdf/bibliotheek/knmipubTR/TR290.pdf.

Overeem, A., Vlemmix, T., van Wijngaarden, F., Mathijssen, T., Veldhuizen, M., Lankamp, B., de Jong, M., and Leijnse, H.: D.T1.1.3 Improvement in Radar Precipitation, https://emfloodresilience.eu/publish/pages/7424/interregradar20231031_d-t1-1-3_v2.pdf, Interreg report EMFloodResilience project, KNMI, De Bilt, 2023b.

Ryzhkov, A., M. Diederich, P. Zhang, and C. Simmer, 2014: Potential Utilization of Specific Attenuation for Rainfall Estimation, Mitigation of Partial Beam Blockage, and Radar Networking. *J. Atmos. Oceanic Technol.*, 31, 599–619, https://doi.org/10.1175/JTECH-D-13-00038.1.

Testud, J., Le Bouar, E., Obligis, E., and Ali-Mehenni, M.: The rain profiling algorithm applied to polarimetric weather radar, J. Atmos. Oceanic Technol., 17, 332–356, https://doi.org/10.1175/1520-0426(2000)017<0332:TRPAAT>2.0.CO;2, 2000.

Wessels, H.R.A., 1972. Metingen van regendruppels te de Bilt. Scientific report. KNMI number: WR-72-06, KNMI, De Bilt (in Dutch).

**RC2**: ['Comment on essd-2025-160'](), Anonymous Referee #2, 11 May 2025

The reviewed paper addresses the practically important topic of operational quantitative radar rainfall estimation, benefiting from the authors' extensive experience in this domain. While the algorithms presented are grounded in well-established statistical and physical concepts and are clearly documented, they are not novel and introduce no fundamentally new elements. More recent literature includes machine learning approaches for radar QPE and geostatistics for radar-raingauge merging, which are not discussed in this work. Nevertheless, the paper's key contribution lies in the accessibility of its data set and the description of the algorithms, which are made publicly available online. This facilitates benchmarking and further development, making the work valuable as a resource. This is a commendable and of great value to the community.

I have some comments regarding the description of the algorithms listed below. From my point of view, the manuscript can be published after consideration of these comments.

**We thank the reviewer for recognizing the value of our work and for the constructive review. Our current real-time radar product will indeed serve as a basis for further product development.**

- The evaluation of the old versus the new QPE algorithm suite is based on data of two different years. It compares the performance of the old suite in 2022 with that of the new suite in 2023. I have two concerns with this approach: a) The comparison is not using the same data. This makes it difficult to compare the algorithms due to interannual variability. I am a bit puzzled by this. Would it be possible to run the two different algorithm suites on the same input data? b) A period of one year is maybe a bit short if one wants to assess the performance of a QPE algorithm suite for all types of precipitation in a given region. Would it be possible to extend the study to 5 years or more?
  **a) Our assumption is indeed that interannual variability will not have a significant effect on the performance evaluation statistics. It is currently not possible to rerun the entire processing chain for the used periods because we do not have an archive of all data going into the QPE products. We start our discussion (Section 5) explaining this point.**
  **b) We agree with the reviewer that extending the analyses to a period of 5 years would increase the robustness of the performance statistics. However, a full year covers all seasons, and we expect that it will be representative for all types of precipitation. Furthermore, extending the period will also mean that there will be more variability in availability and settings of the radars, which could introduce discontinuities in**

**performance. And given the lack of an archive of data relevant for this product (see our reply to point a)) it is not feasible to extend the analyses to a longer period.**

- Machine learning opens new ways to process radar data and generate estimates of rainfall rates and amounts on the ground (QPE). The manuscript makes no mention of this new avenue for QPE. I propose to add a paragraph in the introduction to mention that this new avenue exists and briefly explain why the authors have opted not to use machine learning in the new QPE suite.
  **We agree that machine learning is an important potential candidate to improve QPE. We added to the introduction: "Machine learning is a promising technique to improve radar-based QPE (e.g., Li et al, 2024), as has for instance been demonstrated with a random forest model trained on 4 years of data and evaluated for six case studies in Switzerland (Wolfensberger et al., 2021). While we do not rule out the future use of machine learning in our operational radar products, we choose to follow a more physically-based approach. A disadvantage of machine learning is that it is optimized using a training dataset, and it could hence be vulnerable to changes in radar availability and settings in an unpredictable way. For some processing steps, such as attenuation correction, we expect that this is more robust because it is based on physics and expected to work in a variety of environmental conditions."**

- The authors mention a latency time of about 2 minutes. a) How is latency defined / measured? Is it limited to the processing time of the QPE algorithms? Or does it include all the steps between the measurement at the radar site and the time when the multi-radar QPE product is ready for dissemination to customers? b) Is the latency of 2 minutes the latency in an operational setting averaged over a large period?
  **a) Here, the latency is the time between the timestamp in the filename/metadata (defined as the end of the last scan) and dissemination. So, it includes the signal processing, data transfer from radar site to cloud environment, processing by the radar manufacturer's software, processing of QPE algorithms, and publishing in our public data portal. We clarified this in the text by adding: "Here, the latency is the time between the end of the last scan and dissemination in the public KNMI Data Platform."**
  **b) Yes, this is the average latency over a period of months.**

- As far as I understand the adjustment with gauges is updated every hour at clock hour. There is a risk of discontinuities ("jumps") when one switches from a one-hour time interval with fixed adjustment factors to the next hour. Did the authors observe such jumps? What is the order of magnitude of the jumps (mean and

maximum)? One could avoid the jumps by determining the adjustment factors with hourly aggregations but updated every 5 minutes. Please comment.
**Yes, we observed such jumps as is already stated in the discussion section: "However, this not only results in a less representative adjustment factor field for the current 5 min time interval, but can also cause a sudden change in the 5 min radar precipitation accumulations due to the change of adjustment factor field once an hour (instead of a more gradual change when the adjustment factor field would be computed every 5 min). This is clearly visible in the Supplement (Movie S1) for 04:50–04:55 UTC, 05:50–05:55 UTC and 06:50–06:55 UTC."**

**We do not have exact information on the magnitude of the jumps. We expect that these jumps are small for stratiform rainfall, but expect they can be large for convective rainfall, as shown in the movie. As was already stated in the product outlook section, the real-time product uses a 60 min spatial adjustment factor field being computed every 5 min with a latency of 5–10 min since 9 October 2024. This will avoid the jumps in the current real-time radar product.**

- The gauge network used for bias adjustment has a separation distance in the order of 30 to 40 kilometer. This may be sufficient for bias adjustment in stratiform rainfall with weak spatial gradients and small representativeness errors of the gauges. For convective rainfall with high spatial variability of hourly rainfall amounts the density of the gauges may not be sufficient. Did the authors look into this doing numerical studies? In stratiform rain I am confident that the bias adjustment improves QPE. In convective rainfall with a separation distance between 30 and 40 kilometer things are less obvious. Did the authors evaluate the performance of the bias adjustment specifically for convective rainfall?
**We agree that limited gauge network density can be an issue, especially in convective rainfall. In the discussion section we already write about the limitations of the gauge network density: "The relatively low network density of automatic rain gauges employed in the adjustment and the fact that no gauge data are employed outside the Netherlands, limit the effectiveness to counteract these sources of error."**

**We did not evaluate specifically for either stratiform or convective rainfall, but Figure 10 shows scatter density plots of daily rainfall for winter, dominated by stratiform rainfall, and for summer, dominated by convective rainfall. Although underestimation is clearly lower, the other metrics show worse results in summer: the coefficient of variation of the residuals is**

**clearly lower and the squared Pearson correlation slightly lower. This points to limitations in network density to resolve the spatial structure of rainfall.**

- Some components of the QPE processing suite make use of dual-polarisation capability (for instance, the clutter removal algorithm). This is the way to go as dual-polarisation offers several advantages over single-polarisation QPE. There is however the question of operational robustness. Do the algorithms that use dual-polarisation moments seamlessly fall back to single-polarisation mode in case of issues or failures in the dual-polarisation measurements?
**In case one or more required dual-pol moments are missing, the fuzzy logic and attenuation correction will not be performed. In case only specific differential phase is missing, the modified Kraemer algorithm is applied to correct for attenuation. In case ZDR is missing, the melting layer detection of the VPR detection will not use ZDR but the vertical structure of Z. We expect that usually either all radar variables are present or are missing.**

- Path attenuation is particularly large in the melting layer where snowflakes become coated with liquid water. How does the proposed attenuation correction perform in the melting layer?
**We try to prevent application of attenuation correction in the melting layer as much as possible because specific differential phase will be noisy in the melting layer and hence not suited for reliable attenuation correction. We achieve this by only employing voxels below the forecasted freezing level height. Apart from that, we think that the effect of attenuation due to the melting layer is relatively limited, because its extent is relatively limited and multiple scans from multiple radars are used to obtain the 2D precipitation product.**

- The new QPE algorithm suite includes a module for VPR correction. In the Netherlands with little beam shielding I expect VPR to become relevant at long ranges because of earth curvature and in regions where the lower beams are contaminated by strong clutter, for instance because of wind energy plants and ships. Can the authors provide some quantitative information about the role of VPR correction? What are typical VPR correction factors? What is the percentage of the ground pixels where VPR correction is relevant?
**Based on a full year of data from our two Dutch radars, we found (unpublished results) that VPR correction gives a more than 10 percentage point bias reduction in unadjusted 2D radar accumulations when compared to all manual daily rain gauges from the Netherlands (and even 15 percentage point in winter). This is a large reduction, certainly given the fact that VPR correction is not applied in convective precipitation. The figure below shows the shortest distance to any of these radars and the locations**

of the gauges used for evaluation. Virtually the entire land surface of the Netherlands is within 150 km from a radar, and the majority of gauges within just 100 km. Based on this, we conclude that VPR correction has to be relevant for the majority of ground pixels over the Netherlands and is not only confined to improvement at > 100 km range from a radar.

**(a) Radar and gauge locations**

[Figure]

- I understand that the merging of different scans and the merging of different radars are done separately in two separate steps. What is the reasoning why it is done in two separate steps? From a statistical point of view it would be natural to do it in one single step in a manner that minimizes the final residual uncertainty. Please comment.

**The main reason for not combining all scans from all radars is that radars with more scans would receive more weight in the final product. The fact that there is diversity in the number of scans among the radars used for this product means that this effect can be significant. Our current approach of first combining scans all scans for a given radar and then combining them to a single product significantly limits this effect.**

As far as I understand the advection correction uses 14 images in 5 minutes. That is, there is an image every 21.4 seconds. If using 15 images, the interval is 20 seconds, a somewhat more natural number. Probably I do not correctly understand this technical detail.

> **This is true, to be precise, the 5-minute interval is divided in 14 sub-intervals. This enables to approximately "follow" showers up to a horizontal velocity of 50 m/s. Having images every 20 or 21.4 images does not really matter in the interpolation, because in the end only 5 min images are produced.**

- Uncertainty estimation and the generation of ensembles, two important aspects, are not mentioned in the article. Does the new QPE suite allow to estimate uncertainties and generate ensembles? If not, are there any plans to go in this direction?

  **The new QPE suite already contains an uncertainty estimate. We briefly touch upon this in the future potential developments in the product outlook section: "Improve the quality index that is contained in the product. This quality index field ranges from 0–1 and serves as a proxy for uncertainty in precipitation estimates. Optimizing parameter settings and relationships for the computation of the quality index may yield better estimates of precipitation and its uncertainty. This especially concerns the gauge adjustment, that currently results in quality indices nearing one in the product, which seems not realistic."**

  **We have no concrete plans yet regarding the generation of ensembles for QPE. However, if there is an uncertainty estimate and information on the shape of the distribution and spatial structure of the errors, ensembles could be generated based on this information.**

**Bibliography**

Li, W., Chen, H., & Han, L. (2024). Improving explainability of deep learning for polarimetric radar rainfall estimation. *Geophysical Research Letters*, 51, e2023GL107898. https://doi.org/10.1029/2023GL107898

Wolfensberger, D., Gabella, M., Boscacci, M., Germann, U., and Berne, A.: RainForest: a random forest algorithm for quantitative precipitation estimation over Switzerland, Atmos. Meas. Tech., 14, 3169–3193, https://doi.org/10.5194/amt-14-3169-2021, 2021.

---

## Author Response (AR2)

**RC1**: ['Comment on essd-2025-160'](), Anonymous Referee #1, 4 July 2025

Review of the manuscript "The Dutch real-time gauge-adjusted radar precipitation product" by Overeem et al submitted to ESSD.

The authors clarified all the points and responded to all the comments in a satisfactory way. At this moment the only point that I would still like to raise concerns the double attenuation correction of the German radars. I understand that this is done inadvertently and that recalculating may not be practical at this point, but what is the impact on the final product? Could the authors try to estimate the error caused by this problem? And maybe a note should be made in the metadata about this or somewhere where the users of the dataset can be made aware of the issue.

Apart from this I think the manuscript is improved and should be published at ESSD.

**We thank the reviewer for the positive assessment of our manuscript. Below we address the double attenuation correction on German radar data.**

**We expect that the effect of the double attenuation correction is generally limited because of the following reasons:**

1. **A 1-year evaluation of the modified Kraemer attenuation correction method on Dutch unadjusted radar data (Overeem et al., 2021) shows that underestimation decreases by only 3.4 percentage points on average for 1-h accumulations. It will have a much larger effect for strong convective cases, which are relatively rare: if radar and/or rain gauge 1-h accumulations are larger than 10 mm, a 15.2 percentage point decrease in underestimation is found.**
2. **Using reflectivities that have already been corrected for attenuation in this attenuation correction scheme may lead to extreme overestimates of reflectivity (Hitschfeld and Bordan, 1954). However, the modified Kraemer attenuation correction method ensures that the attenuation correction and the resulting reflectivities are limited (to 10 dB and 59 dB$Z_h$ in our case, respectively; see Overeem et al., 2021). In addition, the reduction of the quality index resulting from the attenuation correction depends on the amount of applied attenuation correction, as shown in the figure below. This implies that the contribution to the composite (and hence the final product) of a pixel that has undergone strong attenuation correction will be small.**

[Figure]

Based on the above reasoning, we added to the results section that already addressed the double attenuation correction (*italic font*):

3. "Another explanation is that the German Weather Service has already corrected their radar data for attenuation, so it has been corrected for attenuation twice (although this also holds for the old product). This was not done on purpose, but only because it was assumed that attenuation correction had not been applied yet. Since its application was not mentioned in the metadata, this indicates the importance of provenance in radar metadata. *This double attenuation correction will likely result in overestimation in case of (strong) convective rainfall. The overall effect is expected to be limited, mainly because the contribution to the composite of pixels that have undergone relatively strong attenuation correction is limited due to the reduction of the quality index associated with attenuation correction (Eq. A4).*"

We assessed the influence of the double attenuation correction for a convective case study employing the horizontally polarized reflectivity factor data from the 0.5-degree elevation scan of the German radar in Essen (top panel, the already attenuation corrected data as obtained from DWD, so without the double attenuation correction). For some azimuths high values for the attenuation

correction are found (bottom panel), implying that the Essen radar data overestimates precipitation by a factor of up to 2.4 (corresponding to a 6 dB overestimate; using the Marshall-Palmer Z-R relation), but that it will receive a much lower weight (0.06 for a 6 dB attenuation correction; see figure above) in the composite for these regions.

[Figure]

Finally, we are planning to update the metadata on the KNMI Data Platform to mention the double attenuation correction.

**Bibliography**

Hitschfeld, W., and J. Bordan, 1954: Errors inherent in the radar measurement of rainfall at attenuating wavelengths. *J. Meteor.*, *11*, 58–67, https://doi.org/10.1175/1520-0469(1954)011<0058:EIITRM>2.0.CO;2.

Overeem, A., H. de Vries, H. Al Sakka, R. Uijlenhoet, and H. Leijnse, 2021: Rainfall-Induced Attenuation Correction for Two Operational Dual-Polarization C-Band Radars in the Netherlands. *J. Atmos. Oceanic Technol.*, 38, 1125–1142, https://doi.org/10.1175/JTECH-D-20-0113.1.